# PRM-PBE: Process Reward Model for Reinforcement Learning in Programming-by-Example

**Yue Fang** [1 2]  **Zhi Jin** [1 2]  **Jie An** [3 4]  **Hongshen Chen** [5]  **Jiangmeng Li** [3 4]  **Xiaohong Chen** [6]  **Naijun Zhan** [1 2 7]

## Abstract

Programming-by-Example (PBE), as a typical few-shot inductive reasoning paradigm, aims to synthesize corresponding algorithms from a set of input-output examples. Although Large Language Models (LLMs) have demonstrated strong program synthesis potential, they still remain ineffective when handling complex PBE tasks. Specifically, LLMs often struggle to accurately grasp the underlying intent of examples, resulting in synthesized programs that either partially satisfy the examples or completely deviate from the target. To address these limitations, we introduce a process-supervised reinforcement learning method that provides fine-grained feedback during the synthesis process, improving the ability of LLMs to capture the intended behavior of provided examples. Firstly, we develop a reasoning tree construction method that is used to build a PBE process supervision dataset. Subsequently, we train a process reward model through preference learning to evaluate the effectiveness of reasoning steps. Finally, we introduce a curriculum learning strategy based on the difficulty of PBE tasks, using Proximal Policy Optimization (PPO) to optimize the model. Experimental results on representative PBE benchmarks show that our approach achieves an average pass rate of 56.61%, significantly outperforming the state-of-the-art baseline by 8.73%.

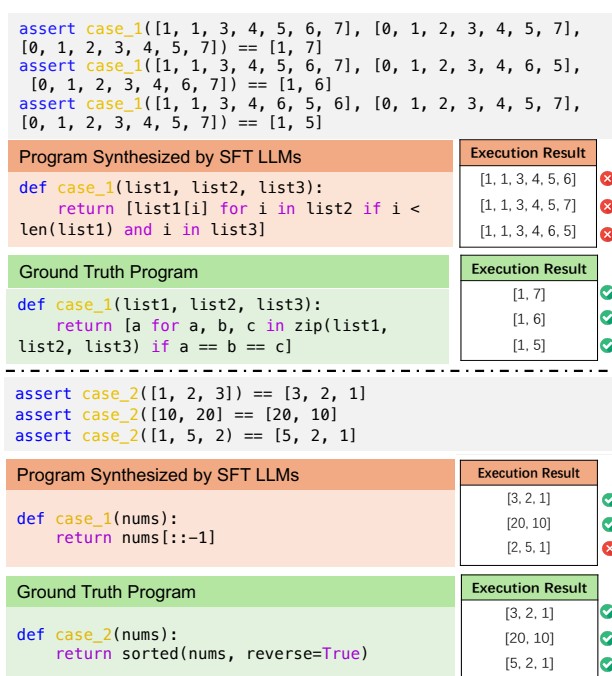

*Figure 1.* Failure cases of the SFT model in capturing latent logical intent from input-output examples.

---

[1]School of Computer Science, Peking University, Beijing, China [2]Key Laboratory of High Confidence Software Technologies (PKU), MOE, China [3]National Key Laboratory of Space Integrated Information System, Institute of Software Chinese Academy of Sciences, Beijing, China [4]University of Chinese Academy of Sciences, Beijing, China [5]JD.com, Beijing, China [6]East China Normal University, Shanghai, China [7]Zhongguancun Laboratory, Beijing, China. Correspondence to: Zhi Jin <zhijin@pku.edu.cn>, Jie An <anjie@iscas.ac.cn>, Yue Fang <y.fang@stu.pku.edu.cn>.

*Proceedings of the $43^{rd}$ International Conference on Machine Learning*, Seoul, South Korea. PMLR 306, 2026. Copyright 2026 by the author(s).

## 1. Introduction

Programming-by-Example (PBE) captures latent intent from a set of input-output examples to automatically synthesize the source code of an underlying function (Lieberman et al., 2000; Gulwani, 2011). As a typical few-shot inductive reasoning paradigm, this task requires models to infer algorithmic logic from limited examples without natural language guidance (Li & Ellis, 2024). However, limited information from examples makes it difficult for models to infer the correct logic, making program synthesis a major challenge.

Early PBE methods rely on Domain-Specific Languages (DSLs) and use symbolic search to synthesize programs that are consistent with given input–output examples. For instance, rule-based synthesis approaches such as Flash-Fill (Gulwani, 2011) synthesize programs by breaking complex problems into smaller parts and solving them incrementally. Machine learning methods (Devlin et al., 2017; Ellis et al., 2023) aim to accelerate the search process by

learning cues from problem descriptions. The rise of Large Language Models (LLMs) has introduced a new paradigm for PBE that no longer relies on predefined DSLs. Current approaches guide models through logical induction using advanced prompting techniques, such as applying Chain-of-Thought to guide program synthesis (Wang et al., 2024b) and incorporating line-level execution feedback (Verbruggen et al., 2025). In addition, SFT approaches are used to train models on input–output examples paired with their corresponding programs to strengthen their ability to synthesize program(Li & Ellis, 2024).

However, existing LLM-based PBE strategies still struggle with complex program synthesis because they primarily rely on input-to-output mapping rather than fine-grained process supervision. As shown in Figure 1, Case 1 requires finding the indices where elements at the same position are equal across multiple lists. Without explicit reasoning steps, the SFT LLM adopts a simple intersection logic. Similarly, Case 2 requires sorting the original sequence in descending order, whereas the SFT LLM incorrectly implements a reversal of the sequence, which only satisfies a subset of the input-output examples. These failures indicate that a lack of correct reasoning over input-output examples results in programs that either deviate completely from the intended logic or satisfy only a subset of examples, failing to provide a comprehensive solution that accounts for all input-output examples.

To address these challenges, we propose integrating a Process Reward Model (PRM) into the reinforcement learning framework for PBE to enhance the model's reasoning alignment with the logical intent of the examples. By explicitly evaluating each intermediate step along the synthesis trajectory, this framework can effectively identify deviation points and prevent the generation of shortcut code that only partially satisfies the constraints. Although the concept of PRM is intuitive, building an effective model and integrating it into reinforcement learning is difficult. The key challenges include accurately assessing the correctness of non-executable reasoning steps in PBE tasks and ensuring stable and effective training when incorporating process-level reward signals.

In this work, we first develop a feedback-guided reasoning tree construction method featuring an external instruction mechanism. This approach builds a high-quality process supervision dataset by introducing natural language descriptions, mitigating the sparsity of positive samples in PBE tasks. Subsequently, candidate nodes are ranked based on the proportion of successor paths that ultimately lead to the correct solution. These rankings define preference signals during PRM training, such that nodes with more successor paths that lead to the correct solution are preferred. A Process Reward Model is then trained using preference learning

to score the effectiveness of intermediate reasoning steps, encouraging the model to prioritize reasoning steps with higher reliability and greater likelihood of success. Finally, we introduce a three-stage curriculum learning strategy organized by failure patterns to progressively regulate task difficulty, while policy updates are performed using Proximal Policy Optimization (PPO) (Schulman et al., 2017) under PRM-guided rewards.

We evaluate our framework on several representative benchmarks, including PROSE (Microsoft, 2025), SyGuS (Alur et al., 2013), Playgol (Cropper, 2016), Lists (Rule et al., 2024) and MBPP (Austin et al., 2021). Experimental results show that our method significantly outperforms state-of-the-art baselines, achieving improvements in pass rates. For instance, using DeepSeek-Coder-V2 as the base model, PRM-PBE achieves an accuracy of 56.61%, marking an 8.73% improvement over the strongest baseline. Furthermore, we find that integrating process rewards enables the model to solve PBE tasks that are challenging for existing baselines. Our method covers 97% of the tasks solved by the SFT baseline and also solves additional tasks that the baseline cannot solve.

In summary, our main contributions are as follows:

- We propose a process-supervised reinforcement learning framework for PBE and a feedback-guided reasoning tree construction method with an external instruction mechanism to generate high-quality process-level supervision data.
- We develop a preference-based training paradigm for the PRM leveraging the success rate of successor paths, and a three-stage curriculum learning strategy structured by failure patterns to improve training efficiency.
- We demonstrate the effectiveness of our approach through experiments on multiple benchmarks, achieving significant improvements in pass rates.

## 2. Related Work

### 2.1. Programming-by-Examples

PBE focuses on synthesizing programs that satisfy desired behaviors from input-output examples (Lieberman et al., 2000; Gulwani, 2011; 2016). Representative systems, such as FlashFill (Gulwani, 2011), primarily operate on DSLs using top-down search and recursive decomposition. To enhance search efficiency, machine learning approaches have been proposed to leverage neural networks to guide symbolic search (Devlin et al., 2017; Ellis et al., 2023). For instance, RobustFill (Devlin et al., 2017) treats synthesis as a sequence-to-sequence mapping for spreadsheet tasks, while DreamCoder (Ellis et al., 2023) facilitates the process by automatically learning a library of reusable components from training problems. Despite their success, these approaches

remain constrained by the limited expressiveness of DSLs. Recently, LLMs have emerged as a new paradigm for PBE that does not rely on DSLs, leveraging techniques such as chain-of-thought reasoning (Wei et al., 2022; Wang et al., 2024b) and execution feedback (Verbruggen et al., 2025) to synthesize programs. To enhance the PBE performance of LLMs, Supervised Fine-Tuning (SFT) is employed to train models on synthesized example-program pairs (Li & Ellis, 2024). However, these methods lack fine-grained guidance for the reasoning steps during the synthesis process. In this paper, we propose a process-supervised method to guide LLMs through each intermediate step of PBE tasks.

## 2.2. Process Supervision

Process Reward Model (PRM) enhances reasoning quality by providing step-by-step reward signals during the generation process, guiding models toward better reasoning paths (Setlur et al., 2024; Zhang et al., 2025). In mathematical reasoning, PRM demonstrates significant success because intermediate steps are highly correlated with the final answer (Luo et al., 2024; Wang et al., 2024a). For instance, Lightman et al. (2023) developed the PRM800K dataset for process supervision, and recent studies focus on automated methods for collecting process-level data to reduce annotation costs (Jiao et al., 2024; Luo et al., 2024). In the area of code generation, researchers utilize the verifiable nature of code to construct PRM datasets and perform process supervision. For example, Dai et al. (2024) utilizes LLMs to complete code prefixes and verify the results through automated tests to build datasets. Similarly, Li et al. (2025) incorporates execution feedback into the PRM to enhance code generation performance. Furthermore, Ye et al. (2025) generates PRM datasets by applying code transformations via a teacher model, subsequently using the PRM to provide fine-grained rewards that guide reinforcement learning. In PBE tasks, induction from input-output examples is often inaccurate as the process often lacks supervision; we introduce process supervision to enhance the ability of LLMs to capture latent intents.

## 3. Methodology

In this section, we present the PRM-PBE framework, as illustrated in Figure 2. First, we construct an automatic process-supervised dataset for PBE tasks. The dataset organizes reasoning processes in a tree structure and incorporates external knowledge guidance to expand reasoning paths effectively. Next, we train a PRM using preference learning, encouraging it to reward more stable reasoning trajectories. Finally, we design a curriculum learning strategy that schedules training data according to the difficulty of the PBE task and employs the Proximal Policy Optimization (PPO) algorithm to optimize the program synthesizer. In

the following, we present the details of each process after a concise formulation of PBE under LLM.

### 3.1. Problem Formulation

For PBE tasks, we consider a set of input-output pairs $D = \{(x_i, y_i)\}_{i=1}^{k}$ which serve as input-output examples. The LLMs are treated as a probabilistic policy conditioned on a prompt $T(D)$ constructed from these examples, used to synthesize a program $P$:

$$P \sim \pi_\theta(\cdot \mid T(D))$$

The objective of the model is to produce a program $P$ that executes correctly on all provided examples, such that:

$$\text{Exec}(P, x_i) = y_i, \quad \forall i = 1, \ldots, k$$

where $\text{Exec}(\cdot, \cdot)$ denotes the output of the program when run on a specific input.

### 3.2. Process-Supervised Dataset Construction

LLMs often perform poorly when performing PBE tasks (Li & Ellis, 2024). This inefficiency leads to sparse positive samples during process supervised dataset construction, reducing the accuracy of the training process reward models. To address this issue, we propose a feedback-guided tree construction method that leverages external instructions to improve program synthesis.

**PBE Reasoning Tree Construction** Each node $s_t$ in the reasoning tree represents a reasoning step in natural language. We first perform logical induction on the input-output examples to obtain the intermediate steps. For any node $s_{t-1}$, the model generates $N$ successor candidates $s_t^{(j)}$ based on the provided examples $E$:

$$s_t^{(j)} = \text{LLM}(E, s_{1:t-1}), \quad j \in \{1, \ldots, N\}$$

Since reasoning nodes cannot be executed directly, we use a post-hoc verification strategy. Validation is performed only after a path reaches a terminal node $s_k$ and is converted into a complete program $C$. The node value $V(s)$ represents the success rate among $N$ sampled successors:

$$V(s) = \frac{1}{N} \sum_{i=1}^{N} \mathbb{I}(s_i \in \mathcal{S}_{success})$$

where $\mathbb{I}(\cdot)$ is an indicator function that equals 1 if a successor $s_i$ leads to a correct solution, and 0 otherwise.

**External Instruction Guidance** Since the reasoning process depends on previous steps, errors made at early stages propagate along the current trajectory, causing all subsequent nodes on that path to be incorrect. To alleviate the

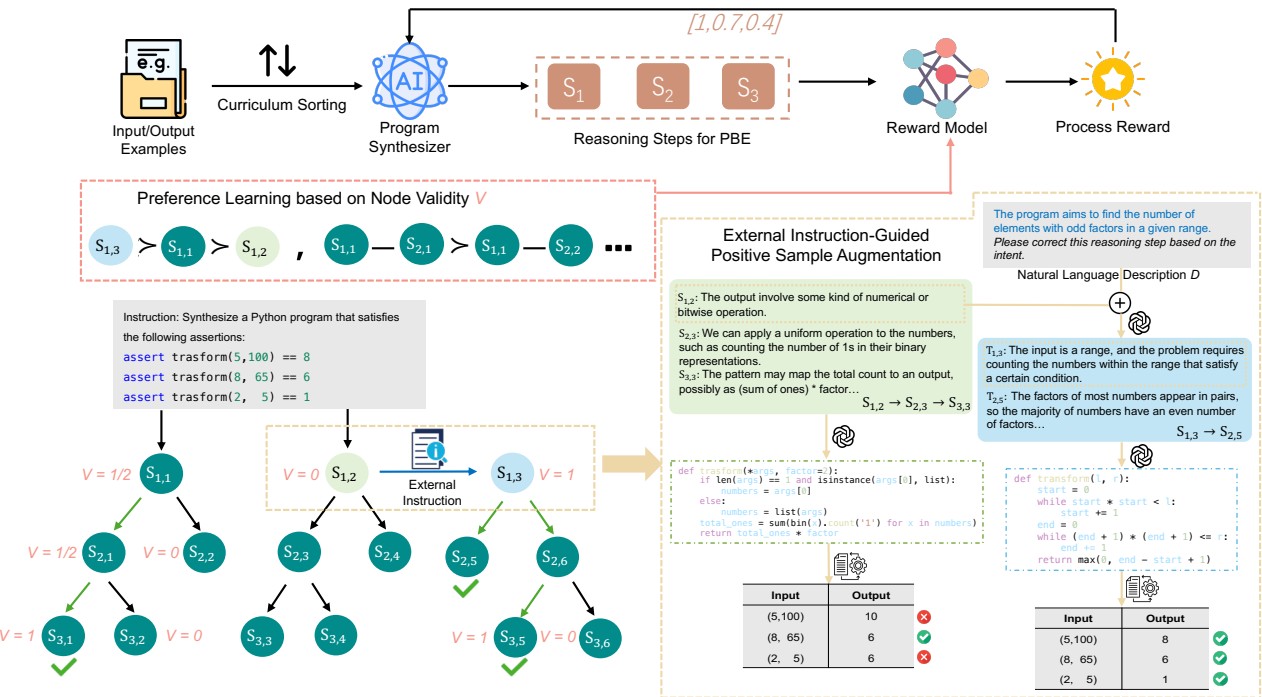

*Figure 2.* Overview of the PRM-PBE framework for process-supervised program synthesis in PBE tasks.

sparsity of positive samples, we adopt a targeted intervention mechanism that begins with logical failure detection. Specifically, after extensive sampling, if all successor nodes of $s_{t-1}$ have zero node values $V(s_t^{(j)})$, this indicates a logical failure in the generation of $s_{t-1}$. In this case, simply increasing the sampling width is ineffective. Once such a failure occurs, the system performs an in-place repair on the faulty node $s_{t-1}$. We retain the original erroneous trajectory and reset the prompt of $s_{t-1}$ by augmenting it with a targeted natural language description $D$:

$$\text{Prompt}(s_{t-1}) \leftarrow \text{Prompt}(s_{t-1}) \oplus D$$

Guided by the description $D$, the model then regenerates the node:

$$s_{t-1}^{\text{new}} = \text{LLM}(D, E, s_{1:t-2})$$

### 3.3. Process Reward Model Training

The objective of PRM is to learn a reward function that accurately estimates the value of a reasoning state $s$, quantifying its potential to lead to a correct program. Specifically, $V(s) = 1$ indicates that the path is definitely on the right track, meaning all subsequent sampling trajectories originating from this node successfully lead to a correct result. A value of $0 < V(s) < 1$ signifies a path with both potential and risk, where the resulting continuations from this point include both successful and failed outcomes. Conversely, $V(s) = 0$ represents a failed trajectory, implying that no correct programs can be generated from this node.

We then employ a pairwise ranking loss to train the model, enabling it to prioritize these partial paths based on the relative magnitude of their node values. For any pair of nodes $(s_i, s_j)$ such that $V(s_i) > V(s_j)$, the loss function is defined as:

$$L(\theta) = -\log \sigma(r_\theta(s_i) - r_\theta(s_j))$$

where $r_\theta(s)$ denotes the reward score assigned by the PRM to the path ending at node $s$. By minimizing the loss function, the PRM learns to distinguish critical decision points and assign higher scores to more reliable reasoning steps.

### 3.4. RL based on Curriculum Learning

**Curriculum Design**  We design a three-stage curriculum learning strategy that gradually introduces different types of PBE training samples, guiding the model from basic executability to reasoning correctness. Stage 1 trains only on PBE problems whose generated programs contain syntax or runtime errors, ensuring that the model can produce programs that execute successfully. Stage 2 selects PBE problems whose generated programs are executable but completely inconsistent with the target specification, encouraging the model to understand the core constraints of the task and form basic reasoning trajectories. Stage 3 further focuses on PBE problems whose synthesized programs pass only a subset of examples, producing correct outputs for some input–output examples while remaining logically incorrect overall. This stage trains the model to distinguish

overall logical correctness from overfitting to limited examples. Through this staged progression, the model gradually acquires the ability to handle complex PBE tasks.

**Reinforcement Learning Training** Based on this curriculum, we train the policy $\pi_\theta$ to maximize the expected reward provided by the PRM. For a given PBE task $x$ and its synthesized reasoning path $y = (s_1, \ldots, s_T)$, the objective function $J(\theta)$ is defined as:

$$J(\theta) = \mathbb{E}_{x \sim \mathcal{D}_c, y \sim \pi_\theta(y|x)} \left[ \sum_{t=1}^{T} \gamma^{T-t} R_\phi(s_t) \right]$$

where $\mathcal{D}_c$ denotes the task distribution of the current curriculum stage, $\gamma$ is the discount factor, and $R_\phi(s_t)$ represents the reward assigned by the PRM.

To update the policy $\pi_\theta$ stably, we employ the Proximal Policy Optimization (PPO) algorithm to maximize the following clipped surrogate objective $L(\theta)$, which serves as a lower bound for $J(\theta)$:

$$L(\theta) = \mathbb{E}_t[\min(\rho_t(\theta)\hat{A}_t, \text{clip}(\rho_t(\theta), 1 - \epsilon, 1 + \epsilon)\hat{A}_t)]$$

where $\rho_t(\theta) = \frac{\pi_\theta(a_t|s_t)}{\pi_{\theta_{old}}(a_t|s_t)}$ is the probability ratio between the current and the old policy, and $\hat{A}_t$ is the advantage estimate derived from the process-level rewards.

# 4. Experiment

## 4.1. Experiment Settings

**Datasets** We conduct experiments on five mainstream PBE benchmarks. PROSE (Microsoft, 2025) is a benchmark with 354 tasks. SyGuS (Alur et al., 2013), derived from the string category of the SyGuS competition, contains 205 cases. Playgol (Cropper, 2016) includes 327 inductive logic programming problems. Lists (Rule et al., 2024) contains 251 list manipulation cases. MBPP (Austin et al., 2021) is adapted for PBE by removing natural language descriptions and retaining only assertions, from which we select 391 problems.

**Evaluation Metrics** Following prior work (Verbruggen et al., 2025), we evaluate PBE performance using the Pass@k (Chen, 2021) metric, which calculates the probability that at least one of the top $k$ generated candidates successfully passes all test cases. For our main results, we report Pass@1 using greedy sampling, generating a single candidate per task.

**Baselines** We evaluate seven advanced LLMs, including three proprietary-source models (GPT-4o, Claude-3.5-Sonnet, and Gemini-1.5-Flash (Team et al., 2024)) and four open-source models (Qwen2.5-Coder (Hui et al., 2024),

DeepSeek-Coder-V2 (Zhu et al., 2024), Llama-3 (Grattafiori et al., 2024), and Qwen3 (Yang et al., 2025)). For the proprietary-source models, we assess their performance under various prompting techniques, including Few-shot (using 5 examples), Chain-of-Thought (CoT) (Wei et al., 2022), where the model first explains its reasoning before generating code; Tree-of-Thought (ToT) (Yao et al., 2023), where the model evaluates each line of code in real time and backtracks if the evaluation is poor to find a better solution; and Within Prompt Search (WPS) (Verbruggen et al., 2025), which integrates search results as feedback into the prompt to guide the model in generating subsequent code. For the open-source models, we evaluate their performance under Few-shot, Supervised Fine-Tuning (SFT) (Li & Ellis, 2024) and our proposed PRM-PBE.

**Implementation Details** We conduct experiments on 8 NVIDIA A800 GPUs, each equipped with 80GB of memory. The framework is implemented using PyTorch (Paszke, 2019) and Huggingface Transformers (Wolf et al., 2020), with LLaMA-Factory (Zheng et al., 2024) serving as the foundation for model customization. In the supervised fine-tuning process, we adopt the data synthesis method proposed in the paper (Li & Ellis, 2024) for data generation For each open-source model, the process reward model and the program synthesis model are the same, both corresponding to the respective open-source model. More details and hyperparameter settings are provided in the Appendix A.

## 4.2. Main Results

According to the results presented in Table 1, we can see that the proposed PRM-PBE model outperforms all baselines across all benchmarks. Compared to the SFT baseline, PRM-PBE achieves significant improvements in the average Pass@1 metric. For instance, on the DeepSeek-Coder-V2 model, PRM-PBE increases the average accuracy from 42.76% to 56.61%, with an absolute improvement of +8.73% compared to the strongest baseline, WPS. This improvement shows high consistency across five different types of test sets, demonstrating the effectiveness of process rewards in PBE tasks.

Furthermore, we observe that within the domain of open-source large language models, specialized code-specific models outperform general-purpose ones. For example, despite its slightly smaller parameter scale, Qwen2.5-Coder surpasses both Llama-3 and Qwen-3 across all metrics. For proprietary-source models, we find that the Claude-3.5-Sonnet model performs well on average, with nearly all top-performing baselines derived from this model. It is also worth noting that the WPS method performs the best in nearly all prompt strategies, although it is still weaker than our proposed PRM-PBE method.

*Table 1.* We compare our proposed PRM-PBE model with baseline methods on five datasets using Pass@1(%) as the evaluation metric. #Param represents the model's parameters, #Avg represents the average value, and the underscore indicates the best performance among all baseline methods. Green subscripts highlight the absolute improvement of our method over the best baseline method.

| Model | Method | #Param. | Lists | Playgol | SyGuS | MBPP | PROSE | #Avg. |
|---|---|---|---|---|---|---|---|---|
| | | | *Proprietary-Source LLMs* | | | | | |
| GPT-4o | Few-shot | – | 28.81 | 50.72 | 52.83 | 32.81 | 41.73 | 41.38 |
| | CoT | | 28.61 | 53.82 | 52.89 | 31.83 | 42.93 | 42.02 |
| | ToT | | 30.02 | 52.83 | 54.81 | 33.72 | 42.83 | 42.84 |
| | WPS | | 32.89 | 58.82 | 57.33 | 38.71 | 49.72 | 47.49 |
| Claude-3.5-Sonnet | Few-shot | – | 29.32 | 51.82 | 53.82 | 34.71 | 43.62 | 42.66 |
| | CoT | | 30.01 | 53.81 | 52.83 | 39.91 | 44.83 | 44.28 |
| | ToT | | 31.79 | 55.83 | 54.73 | 38.62 | 47.82 | 45.76 |
| | WPS | | 33.04 | 59.71 | 58.23 | 37.62 | 50.81 | 47.88 |
| Gemini-1.5-Flash | Few-shot | – | 30.12 | 49.58 | 52.13 | 35.81 | 45.83 | 42.69 |
| | CoT | | 28.76 | 50.82 | 55.73 | 36.72 | 45.89 | 43.58 |
| | ToT | | 29.83 | 52.73 | 54.27 | 38.72 | 47.32 | 44.57 |
| | WPS | | 29.96 | 58.72 | 59.72 | 39.75 | 49.93 | 47.62 |
| | | | *Open-Source LLMs* | | | | | |
| Qwen2.5-Coder | Few-shot | 7.6B | 20.83 | 48.72 | 46.63 | 30.71 | 39.82 | 37.34 |
| | SFT | | 27.73 | 50.37 | 48.72 | 33.72 | 42.82 | 40.67 |
| | **PRM-PBE (Ours)** | | **40.33** +7.29 | **65.32** +5.61 | **65.32** +5.60 | **47.12** +7.21 | **57.22** +6.41 | **55.06** +7.18 |
| DeepSeek-Coder-V2 | Few-shot | 15.7B | 23.73 | 50.12 | 50.89 | 30.92 | 40.81 | 39.29 |
| | SFT | | 28.73 | 53.72 | 52.93 | 34.72 | 43.72 | 42.76 |
| | **PRM-PBE (Ours)** | | **41.52** +8.48 | **66.93** +7.22 | **66.76** +7.04 | **48.72** +8.81 | **59.13** +8.32 | **56.61** +8.73 |
| Llama-3 | Few-shot | 8B | 18.72 | 45.82 | 47.72 | 28.71 | 38.72 | 35.94 |
| | SFT | | 25.83 | 48.61 | 49.62 | 30.71 | 41.73 | 39.30 |
| | **PRM-PBE (Ours)** | | **39.72** +6.68 | **64.71** +5.00 | **64.83** +5.11 | **44.81** +4.90 | **56.83** +6.02 | **54.18** +6.30 |
| Qwen-3 | Few-shot | 8B | 19.33 | 43.73 | 45.31 | 29.63 | 37.92 | 35.18 |
| | SFT | | 26.83 | 46.72 | 46.62 | 31.82 | 40.72 | 38.54 |
| | **PRM-PBE (Ours)** | | **39.32** +6.28 | **64.74** +5.03 | **65.02** +5.30 | **45.32** +5.41 | **57.41** +6.60 | **54.36** +6.48 |

Regarding prompt strategies, while both CoT and ToT are based on the inductive reasoning approach of input-output examples, similar to our method, their performance is constrained by the reasoning limitations of the underlying LLMs. This further validates that for complex PBE tasks, providing explicit and verifiable reasoning significantly improves the accuracy of program synthesis.

In conclusion, the experimental results across five diverse benchmarks, as well as the application of PRM-PBE across four different backbone LLMs, validate the effectiveness and robustness of our methods.

## 5. Analysis

### 5.1. Ablation Study

To validate the contribution of each component in the PRM-PBE framework to the final performance, we conduct ablation experiments using two base models: Qwen-3 (8B) and DeepSeek-Coder-V2. As shown in Table 2. We evaluate the Pass@1 metric on three representative datasets: Lists, MBPP, and PROSE. The w/o External Instruction variant

*Table 2.* Ablation study of PRM-PBE. We report Pass@1 (%) on three representative datasets and the average performance. Subscripts indicate the performance drop compared to the Full model.

| Model | Lists | MBPP | PROSE | Average |
|---|---|---|---|---|
| *Qwen-3 (8B)* | | | | |
| **PRM-PBE (Full)** | **39.32** | **45.32** | **57.41** | **47.35** |
| – w/o External Intervention | $34.51_{-4.81}$ | $40.62_{-4.70}$ | $52.53_{-4.88}$ | $42.55_{-4.80}$ |
| – w/o Preference Learning | $36.42_{-2.90}$ | $42.51_{-2.81}$ | $54.72_{-2.69}$ | $44.55_{-2.80}$ |
| – w/o Curriculum Learning | $36.85_{-2.47}$ | $43.05_{-2.27}$ | $54.91_{-2.50}$ | $44.94_{-2.41}$ |
| *DeepSeek-Coder-V2* | | | | |
| **PRM-PBE (Full)** | **41.52** | **48.72** | **59.13** | **49.79** |
| – w/o External Intervention | $36.75_{-4.77}$ | $43.82_{-4.90}$ | $54.31_{-4.82}$ | $44.96_{-4.83}$ |
| – w/o Preference Learning | $38.71_{-2.81}$ | $45.92_{-2.80}$ | $56.43_{-2.70}$ | $47.02_{-2.77}$ |
| – w/o Curriculum Learning | $39.22_{-2.30}$ | $46.51_{-2.21}$ | $56.91_{-2.22}$ | $47.55_{-2.24}$ |

removes natural language corrections during reasoning tree construction; the system increases sampling width instead of repairing logic. The w/o Preference Learning variant replaces pairwise ranking with binary supervised learning, where only paths leading to entirely correct programs are labeled positive. Lastly, w/o Curriculum Learning optimizes the policy using the final reward without task scheduling. Detailed ablation results for other LLMs and performance on other datasets can be found in the Appendix C.

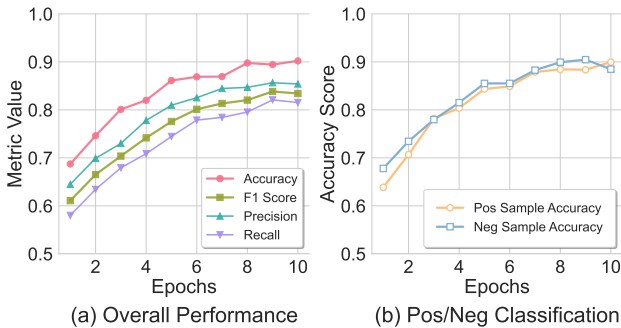

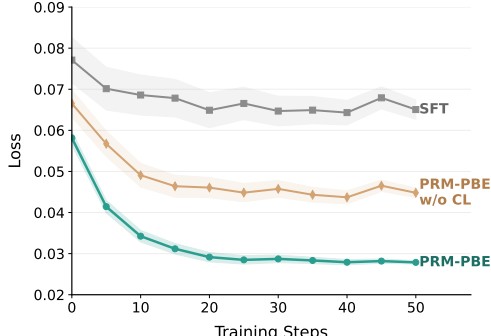

*Figure 3.* Performance evaluation of the process reward model based on DeepSeek-Coder-V2 for the MBPP dataset.

*Figure 4.* Comparison of training loss curves for different supervision methods.

According to the results in Table 2, we draw the following conclusions: The experimental data indicate that the removal of external guidance has the most significant impact on performance, causing an average performance drop of about 4.8%. This proves that in PBE tasks, the effectiveness of PRM is highly dependent on the quality of dataset construction. After removing preference learning, the model's performance drops by about 2.8%, with this phenomenon being particularly evident in the logically complex Lists dataset. This proves the effectiveness of preference learning in PRM-PBE. The removal of curriculum learning leads to a performance degradation of about 2.3%. This validates that the multi-stage curriculum learning we designed enables the model to robustly master the PBE tasks.

### 5.2. Accuracy of Reward Model

To validate the effectiveness of the PRM in evaluating reasoning steps, we record the model classification metrics on the process supervision dataset during training. As shown in Figure 3 (a), PRM performance metrics steadily improve with the increase in training epochs. By the 10th epoch, the model accuracy reaches approximately 0.90, with the F1 score, precision, and recall all remaining at high levels between 0.80 and 0.90, indicating that PRM accurately assesses the quality of the reasoning path.

Additionally, Figure 3 (b) presents the model accuracy for classifying positive and negative samples. The experimental results show that the classification accuracy for both positive and negative samples increases throughout the training process, stabilizing at around 0.88. This not only demonstrates the model accuracy in identifying valid reasoning steps but also reflects its high sensitivity in capturing error signals that lead to logical deviations.

### 5.3. Training Efficiency

To evaluate the optimization stability of our method, we monitor the training loss of PRM-PBE using the DeepSeek-Coder-V2 model on the MBPP dataset, and compare it with

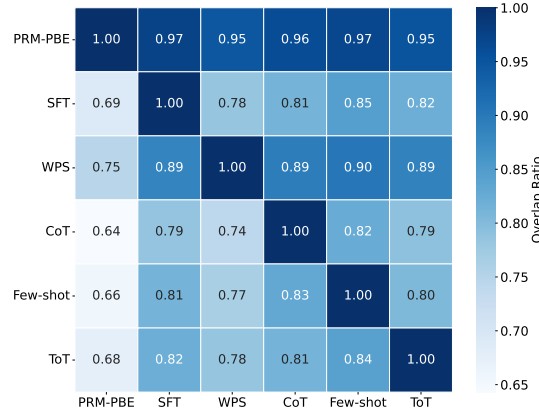

*Figure 5.* Heatmap of pairwise problem solving overlap across all benchmarks.

the SFT model and w/o CL (curriculum learning).

As shown in Figure 4, the loss reduction of the PRM method is steeper than that of the SFT baseline model. This indicates that the dense step-level feedback provided by PRM better optimizes the model. Moreover, the gap between PRM-PBE and the w/o CL variant highlights the necessity of our three-stage training strategy. Without curriculum learning, the model's loss exhibits larger fluctuations. In summary, the integration of process rewards and curriculum learning enhances the stability of the optimization process.

### 5.4. Problem Solving Overlap

To analyze the differences in problem-solving abilities between PRM-PBE and existing methods, we perform a set of overlap analysis between models across all benchmarks. We select six representative methods for comparison: our method PRM-PBE, the SFT baseline based on DeepSeek-Coder-V2, and variations of GPT-4o under different prompting strategies (including WPS, CoT, Few-shot, and ToT). For each pair of models $(M_i, M_j)$, we calculate the overlap rate, which is the proportion of problems solved by $M_i$ that

*Table 3.* Error statistics across different methods on the MBPP dataset. We categorize failures into Execution Errors, Logical Deviations, and Partial Matches.

| Model | Method | Execution Error | Logical Deviation | Partial Match |
|---|---|---|---|---|
| GPT-4o | Few-shot | 29 | 42 | 40 |
| | CoT | 22 | 47 | 31 |
| | ToT | 25 | 40 | 35 |
| | WPS | 21 | 39 | 29 |
| DeepSeek-Coder-V2 | Few-shot | 33 | 37 | 32 |
| | SFT | 24 | 43 | 33 |
| | **PRM-PBE** | **17** | **21** | **20** |

are also solved by $M_j$.

As shown in Figure 5, PRM-PBE demonstrates extremely high coverage across all baseline methods. It successfully covers 97% of the problem set solved by the SFT model, and 95% of the strong baseline WPS. In contrast, baseline methods exhibit much lower coverage of the PRM-PBE solution set, generally ranging from 64% to 75%. This indicates that for some complex PBE tasks, our process-supervised method exhibits stronger problem-solving ability, while even closed-source models using advanced prompting strategies like CoT, ToT, and WPS are unable to address these problems.

### 5.5. Error Analysis

We conduct a statistical analysis of erroneous programs generated by different baseline models on the MBPP dataset. Specifically, we randomly sample and analyze 150 synthesis-failure cases across two backbone categories: closed-source models (GPT-4o with Few-shot, WPS, CoT, and ToT) and open-source models (DeepSeek-Coder-V2 with SFT and our PRM-PBE). Based on the analysis of these failure cases, we categorize erroneous programs into three main types: execution errors, logical deviations, and partial matches, where the latter two correspond to cases in which the program outputs are completely mismatched with the examples or only partially matched, respectively.

As illustrated in Table 3, we make the following observations. PRM-PBE produces only 21 logical deviation errors, which is fewer than the other methods, indicating that process-level supervision helps the model better capture the underlying global logic in PBE tasks. In the partial match category, PRM-PBE also yields fewer errors (20) than both the SFT baseline (33) and GPT-4o with WPS prompting (29), suggesting that our method effectively suppresses shortcut-style program generation and encourages reasoning over all input–output examples. Moreover, even when compared to GPT-4o using its strongest prompting strategy, PRM-PBE outperforms in the execution error cat-

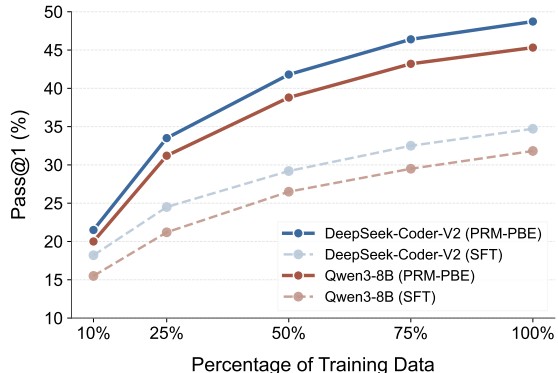

*Figure 6.* Comparison of Pass@1 (%) performance between PRM-PBE and the SFT baseline on MBPP dataset as the percentage of training data increases.

*Table 4.* We compare PRM-PBE with four ORM variants, all built on DeepSeek-Coder-V2 as the base model. We report Pass@1 (%) on five PBE benchmarks. #Avg. denotes the average across datasets, and the best score in each column is shown in **bold**.

| Method | Lists | Playgol | SyGuS | MBPP | PROSE | #Avg. |
|---|---|---|---|---|---|---|
| ORM-Binary (w/o $D$) | 29.73 | 56.28 | 56.45 | 35.82 | 44.16 | 44.49 |
| ORM-Binary | 32.12 | 58.62 | 58.88 | 38.21 | 46.72 | 46.91 |
| ORM-Fractional | 33.89 | 60.47 | 60.35 | 40.28 | 49.14 | 48.83 |
| ORM-Learned | 34.54 | 60.83 | 60.47 | 40.95 | 49.68 | 49.29 |
| **PRM-PBE (ours)** | **41.52** | **66.93** | **66.76** | **48.72** | **59.13** | **56.61** |

egory, showing the benefit of incorporating process-level supervision during training rather than relying solely on inference-time search and feedback.

### 5.6. Scaling Analysis

To further investigate the performance of the PRM-PBE framework under different data scales, we select DeepSeek-Coder-V2 and Qwen3-8B as the base models and compare them with the SFT baseline method. In the experiment, the size of the training data gradually increases from 10% to 100% of the total amount. As shown in Figure 6, with the increase in the percentage of training data, the Pass@1 accuracy of all models shows a significant growth trend. At the same data scale, PRM-PBE consistently outperforms the SFT baseline. For example, PRM-PBE already outperforms the SFT model trained with 100% data when using only 50% of the training data. While model performance continues to improve as the data size increases, we also observe that as the data scale further expands, the rate of performance improvement gradually slows down.

### 5.7. PRM vs. ORM: Effect of Reward Granularity

To disentangle the contribution of process-level supervision from that of the reinforcement learning framework, we compare PRM-PBE with a family of outcome reward models (ORM) under a controlled setting that isolates reward gran-

ularity from other factors. All variants share the same base model (DeepSeek-Coder-V2) and the same PPO pipeline. ORM-Binary (w/o $D$) is a standard RLVR-based baseline that returns $1$ when the program passes all test cases and $0$ otherwise, with neither reasoning trees nor external instruction $D$. ORM-Binary uses the same binary execution reward but on our $D$-augmented data. ORM-Fractional assigns partial credit $R = k_{\text{pass}}/k_{\text{total}}$ proportional to the fraction of passed examples. ORM-Learned is trained with the Bradley–Terry objective on complete programs ranked by their pass count. PRM-PBE provides process reward $R_\phi(s_t)$ on intermediate reasoning states.

As shown in Table 4, PRM-PBE achieves the highest Pass@1 on all five PBE benchmarks. Even the strongest ORM variant, ORM-Learned, reaches only $49.29$ on average, which is still surpassed by the $56.61$ of PRM-PBE. Among the remaining ORM variants, ORM-Binary (w/o $D$), ORM-Binary, and ORM-Fractional reach average scores of $44.49$, $46.91$, and $48.83$ respectively, all of which remain below PRM-PBE. These results indicate that in reinforcement learning for PBE tasks, the process reward signal of PRM-PBE is more effective than outcome-level reward used by ORM methods.

## 6. Conclusion

In this paper, we present PRM-PBE, a process-supervised reinforcement learning framework designed to enhance the performance of LLMs on PBE tasks. Our approach leverages feedback-guided reasoning trees to construct datasets and a Process Reward Model trained via preference learning, guided by a three-stage curriculum from basic syntax to complex patterns. Experiments on five benchmarks demonstrate that PRM-PBE significantly outperforms baselines. In future work, we aim to extend the framework to more diverse programming paradigms beyond string and list manipulation, such as graphics processing.

## Acknowledgements

This work is supported by the National Natural Science Foundation of China under Grant Nos. 62192731, 62192732, 62192730, W2511064, and 92582203, the CAS Project for Young Scientists in Basic Research under Grant No. YSBR-123, the ISCAS Basic Research under Grant No. ISCAS-JCZD-202406.

## Impact Statement

PRM-PBE relies on natural language descriptions $D$ that are derived from the reference programs in the dataset. These descriptions are used only during training data construction and are not required at inference time, but they introduce oracle information that goes beyond the input-output examples available at test time. Because $D$ is used to repair dead-end nodes during reasoning tree construction, it plays a non-trivial role in producing positive supervision signals, and therefore contributes to part of the performance gains reported in our experiments. The framework is thus most applicable when reference programs or equivalent descriptive information are available during training data construction.

In addition, PRM-PBE involves reasoning tree construction, PRM training, and curriculum-based PPO optimization, which require more computation than standard SFT. Improving the efficiency of process data construction and reducing the reliance on reference-derived descriptions remain useful directions for future work.

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

# A. Implementation Details

The learning rate is set to $5 \times 10^{-6}$ for the policy model during the PPO stage and $1 \times 10^{-5}$ for the PRM during the preference learning phase. We utilize a global batch size of 128, implemented via gradient accumulation to adapt to the hardware environment consisting of 8 NVIDIA A800 GPUs. For the PPO optimization , we employ a clipping range of $\epsilon = 0.2$ , a discount factor of $\gamma = 0.95$ , and Generalized Advantage Estimation (GAE) for computing advantage estimates. Regarding the architectural details of the PRM, a linear layer (Value Head) is appended to the base model to map the hidden states of the final layer into scalar reward scores.

To construct a robust training benchmark, we curated 30 seed input-output (I/O) examples and their corresponding reference programs for each of the five PBE tasks evaluated in this study. Every seed sample underwent rigorous manual verification to ensure that the programs correctly implemented the logic required by their respective I/O examples. Following (Li & Ellis, 2024), we utilized DeepSeek-V3 to synthesize diverse program logic and associated I/O pairs, expanding the initial seed set to a total of 8,000 synthetic training pairs per task category. To guarantee functional correctness, each synthesized program was executed on the generated inputs to verify that the actual output matched the expected result, and any program failing this verification or encountering runtime errors was discarded. Finally, to ensure the conciseness of the supervision signals, five students with extensive programming experience manually refined the code to eliminate logical redundancies and standardize the implementation style.

# B. Prompts

---

**Prompt for Program Synthesis**

You are an expert AI programmer. Your task is to perform logical induction based on provided examples to synthesize a Python program.

Input-Output Examples: {{input_output_examples}}

Constraints:
1. Start by describing the latent logical intent of these examples in natural language.
2. Implement a Python function solution that executes correctly on all examples.
3. Avoid logical shortcuts or overfitting to only a subset of examples.

---

*Figure 7.* Prompt for program synthesis based on input-output examples.

---

**Prompt for PRM Dataset Construction**

You are a reasoning assistant for Programming-by-Example. You generate step-by-step reasoning nodes to build a process-level supervision dataset.

User Prompt Context: Input-Output Examples: {{target_examples}} Current Path: {{previous_steps}}

Instruction (Normal Generation): Based on the examples and the current path, generate the next logical reasoning step in natural language.

Instruction (External Instruction-Guided Repair): Note: A logical failure was detected in the previous trajectory. To facilitate the generation of a high-quality positive sample, please follow this additional external instruction to reset and repair the logic: External Instruction: {{external_instruction_D}}

Requirement: Ensure the generated node leads toward a terminal program that fulfills the intent of input-output examples.

---

*Figure 8.* Prompt for PRM dataset construction.

The prompts for LLMs to conduct PBE task and PRM dataset construction are shown in Figure 7 and Figure 8.

## C. Comprehensive Ablation Study

The results comprehensive ablation study are shown in Figure 5. For the DeepSeek-Coder-V2 model, the removal of External Intervention, Preference Learning, and Curriculum Learning leads to average Pass@1 decreases of 4.83%, 2.78%, and 2.27%, respectively. Similarly, for the Qwen 3-8B model, the absence of these same components results in average performance drops of 4.80%, 2.80%, and 2.41%. These numerical findings are consistent with the conclusions of the ablation study presented in the main paper.

*Table 5.* Comprehensive ablation study of PRM-PBE on all five benchmarks. We report Pass@1 (%) for both **DeepSeek-Coder-V2** and **Qwen 3-8B**. Subscripts denote the absolute performance drop compared to the Full model.

| Method | Lists | Playgol | SyGuS | MBPP | PROSE | Average |
|---|---|---|---|---|---|---|
| *Model: DeepSeek-Coder-V2 (15.7B)* | | | | | | |
| **PRM-PBE (Full)** | **41.52** | **66.93** | **66.76** | **48.72** | **59.13** | **56.61** |
| w/o External Intervention | $36.75_{-4.77}$ | $62.10_{-4.83}$ | $61.90_{-4.86}$ | $43.82_{-4.90}$ | $54.31_{-4.82}$ | $51.78_{-4.83}$ |
| w/o Preference Learning | $38.71_{-2.81}$ | $64.15_{-2.78}$ | $63.95_{-2.81}$ | $45.92_{-2.80}$ | $56.43_{-2.70}$ | $53.83_{-2.78}$ |
| w/o Curriculum Learning | $39.22_{-2.30}$ | $64.60_{-2.33}$ | $64.45_{-2.31}$ | $46.51_{-2.21}$ | $56.91_{-2.22}$ | $54.34_{-2.27}$ |
| *Model: Qwen 3-8B* | | | | | | |
| **PRM-PBE (Full)** | **39.32** | **64.74** | **65.02** | **45.32** | **57.41** | **54.36** |
| w/o External Intervention | $34.51_{-4.81}$ | $59.93_{-4.81}$ | $60.19_{-4.83}$ | $40.62_{-4.70}$ | $52.55_{-4.86}$ | $49.56_{-4.80}$ |
| w/o Preference Learning | $36.42_{-2.90}$ | $61.94_{-2.80}$ | $62.22_{-2.80}$ | $42.51_{-2.81}$ | $54.72_{-2.69}$ | $51.56_{-2.80}$ |
| w/o Curriculum Learning | $36.85_{-2.47}$ | $62.34_{-2.40}$ | $62.62_{-2.40}$ | $43.05_{-2.27}$ | $54.91_{-2.50}$ | $51.95_{-2.41}$ |

