# OpenReview forum: "PRM-PBE: Process Reward Model for Reinforcement Learning in Programming-by-Example"
_ICML.cc/2026/Conference — ICML 2026 regular_

### Official Review · Reviewer_VczE · 2026-02-26

**Soundness:** 2
**Presentation:** 3
**Significance:** 2
**Originality:** 2
**Overall Recommendation:** 3
**Confidence:** 3

**Summary:**

The paper proposes PRM-PBE, a process-supervised reinforcement learning framework for Programming-by-Example (PBE) tasks. To address the issue of LLMs failing to capture latent logical intents from limited examples, the authors develop a method to construct a process-level supervision dataset using reasoning trees guided by external instructions. A Process Reward Model (PRM) is then trained via preference learning to evaluate intermediate reasoning steps. Finally, the policy model is optimized using PPO, guided by the PRM and a three-stage curriculum learning strategy based on task difficulty. Experimental results on five benchmarks demonstrate that PRM-PBE outperforms standard SFT and various prompt-based baselines (e.g., CoT, ToT, WPS).

**Compliance With Llm Reviewing Policy:**

Affirmed.

**Final Justification:**

Although the authors' response has partially resolved my concerns on the contribution of the paper, the paper itself needs to be re-established to clear the contribution and experiments. I remain my scores.

**Key Questions For Authors:**

See weaknesses.

**Limitations:**

no (not adequately discussed)

**Strengths And Weaknesses:**

Strengths:
1. Relevant problem and clear motivation. PBE is a challenging setting (no problem statement, only examples), and shortcut/partial-match behaviors are a real issue. Using process-level supervision to encourage better “intent inference” is well-motivated.
2. End-to-end pipeline with data construction + PRM + RL. The paper provides a full recipe: constructing process-supervision data (reasoning trees), training PRM via preferences, and then optimizing the policy with PPO under a curriculum. The inclusion of ablations is helpful for understanding which modules matter.

Weaknesses:
1. **Core contribution and novelty are not sufficiently isolated.** The paper’s improvements appear to come from combining (i) additional training data / interventions during tree construction, (ii) PRM training via preference ranking, and (iii) PPO with curriculum. However, it remains unclear which part is the key novel ingredient. The PRM training objective is largely standard preference ranking, and the RL algorithm (PPO) is standard; without stronger comparisons (e.g., alternative PRMs/ORMs, or standard RLVR baselines), it is difficult to attribute gains specifically to the proposed PRM approach.
2. **Unclear generation and cost of “external instruction” used to correct early logic.** A major component appears to be injecting external instruction when branches fail, and ablations indicate this contributes substantially. However, the paper does not clearly specify how these instructions are generated (human-written vs model-generated vs derived from ground-truth programs), what information they use, and what computational/human cost they require. This affects both reproducibility and whether the method changes the task assumptions (e.g., introducing an implicit “problem statement” through instructions).
3. **Limited evidence that PRM truly reduces “partial match” shortcut programs.** While the paper includes an error-type breakdown, the claim that PRM reduces partial matches would be more convincing with targeted statistics (e.g., fraction of solutions passing k-of-n examples), plus qualitative case studies showing how PRM steers reasoning away from plausible-but-wrong hypotheses.
4. **Missing baseline: standard non-PRM RLVR / outcome-reward RL.** The comparisons mainly show that “RL + extra supervision” can outperform SFT. However, to support the claim that PRM is the key driver, the paper should compare to a standard RLVR baseline that uses only outcome reward (e.g., program passes all examples) without PRM shaping, under similar compute/budget. Without this, it is hard to tell whether improvements are due to PRM specifically or simply due to RL/post-training.

---

> ### Author Rebuttal · Authors · 2026-03-30
>
> We deeply appreciate Reviewer for the insightful comments. Your recognition of the motivation of our proposed PRM-PBE framework has greatly encouraged us.
> We believe our response has addressed most of your concerns and we respectfully ask if you're comfortable to re-consider the rating in light of our efforts and dedication. Below, we provide detailed responses to all the comments:
>
> For W1 and W4:
>
> We have supplemented the ORM RL experiment using the following method: specifically, if the generated complete program $P$ passes all test cases (i.e., $Exec(P, x_i) = y_i$), the reward $R = 1$; otherwise, $R = 0$, and reinforcement learning is performed using the same PPO pipeline. The experimental results are as follows.
>
> | Model | Method | Lists | Playgol | SyGuS | MBPP | PROSE | #Avg. |
> |------|--------|-------|---------|-------|------|-------|------|
> | Qwen2.5-Coder | ORM | 31.09 | 57.92 | 57.72 | 37.81 | 45.32 | 45.97 |
> | Qwen2.5-Coder | PRM-PBE | 40.33 | 65.32 | 65.32 | 47.12 | 57.22 | 55.06 |
> | DeepSeek-Coder-V2 | ORM | 32.12 | 58.62 | 58.88 | 38.21 | 46.72 | 46.91 |
> | DeepSeek-Coder-V2 | PRM-PBE | 41.52 | 66.93 | 66.76 | 48.72 | 59.13 | 56.61 |
> | Llama-3 | ORM | 30.72 | 57.68 | 56.93 | 36.62 | 42.32 | 44.85 |
> | Llama-3 | PRM-PBE | 39.72 | 64.71 | 64.83 | 44.81 | 56.83 | 54.18 |
> | Qwen-3 | ORM | 30.21 | 57.31 | 57.01 | 36.77 | 43.18 | 44.90 |
> | Qwen-3 | PRM-PBE  | 39.32 | 64.75 | 65.02 | 45.32 | 57.41 | 54.36 |
>
> It is observed that the improvement brought by PRM is far superior to that of simply applying ORM. Taking DeepSeek-Coder-V2 as an example, PRM-PBE achieves 56.61% accuracy, outperforming ORM method at 46.91% by 9.70%.
> This strongly proves that fine-grained process-level feedback is the key to the performance leap, rather than just the effect of RL training.
>
> For W2:
>
> D refers to the natural language descriptions corresponding to the programs originally included in the dataset. The core reason for introducing these descriptions is the extreme scarcity of positive samples in complex PBE tasks.
>
> However, the performance improvement of the PRM-PBE stems from the process-level reward paradigm, rather than being driven by the oracle information in D. In supplemented experiments, we train an ORM baseline model using the same dataset augmented with D. Under the same training data conditions, the ORM model achieves an accuracy of 46.91%, while PRM-PBE reaches 56.61%. This gap suggests that process-level supervision can extract more informative signals from the data than outcome-level supervision.
>
> For W3:
>
> In response to your suggestion, we have supplemented new statistical data, namely the proportion of programs that pass only part of the test cases. From this table, it can be clearly observed that the proportion of programs passing only part of the test cases is the lowest for the method based on PRM-PBE, which strongly demonstrates that PRM can effectively reduce partial match shortcut programs.
>
> | Model | Method | 0 < k/n < 1 |
> |-|-|-|
> | Claude-3.5-Sonnet | Few-shot | 22.9 |
> | Claude-3.5-Sonnet | CoT | 20.2 |
> | Claude-3.5-Sonnet | ToT | 23.4 |
> | Claude-3.5-Sonnet | WPS | 18.7 |
> | DeepSeek-Coder-V2 | Few-shot | 21.2 |
> | DeepSeek-Coder-V2 | SFT | 20.5 |
> | DeepSeek-Coder-V2 | PRM-PBE | 12.3 |
>
> In response to your suggestion, we have supplemented new statistical data, namely the proportion of programs that pass only part of the test cases. From this table, it can be clearly observed that the proportion of programs passing only part of the test cases is the lowest for the method based on PRM-PBE, which strongly demonstrates that PRM can effectively reduce partial match shortcut programs.
>
> To intuitively demonstrate the error-correcting ability of PRM, we selected a typical list processing task with **intent ambiguity**:
>
> - Example 1: `[10, 20] -> [20, 10]`
> - Example 2: `[3, 8] -> [8, 3]`
> - Example 3: `[1, 5, 2] -> [2, 5, 1]`
>
> When processing the first two examples, the model is prone to generating a **plausible but incorrect** shortcut assumption: "The task objective is to sort the list in **descending order** by numerical value." This logic can perfectly pass the first two examples, but it will generate the incorrect result `[5, 2, 1]` when faced with the test case `[1, 5, 2]`.
>
> By assigning different reward scores at semantic decision points, PRM guides the model to avoid the above trap. The following table shows the difference in PRM scores between the two competing paths:
>
> | Step | Path A (Incorrect "Descending Order") | PRM | Path B (Correct "Reversal") | PRM |
> |------|----------------------------------------|-----|-------------------------------|-----|
> | Step 1 | First output element > second | +0.65 | Output order is reverse of input | +0.78 |
> | Step 2 | Sort descending by value | -0.82 | Reverse element positions | +0.74 |
> | Step 3 | sort(reverse=True) | -0.88 | index-based reversal | +0.78 |
> | Step 4 | sorted(input_list, reverse=True) | -0.95 | input_list[::-1] | +0.98 |

---

> > ### Author Rebuttal · Reviewer_VczE · 2026-04-02
> >
> > I thank the authors for their response. Based on the new evidence provided, it appears that the primary driver of the performance gains is the introduction of the PRM. Given that integrating PRMs into RL for math and code generation is already well-explored in the literature, could you clarify the core novelties of your approach and distinguish it from prior work, beyond simply applying it to the PBE setting?

---

> > > ### Author Response · Authors · 2026-04-06
> > >
> > > We thank the reviewer for this important question. We would like to clarify that our contribution goes beyond simply applying PRM to the PBE setting. Below we highlight the key novelties and distinctions from prior work. We would be grateful if you could re-evaluate our work in light of these clarifications.
> > >
> > > ### 1. Unique Challenges in PBE that Require Novel Solutions
> > >
> > > Unlike math reasoning or general code generation, PBE presents distinct challenges:
> > >
> > > - **Extreme sparsity of positive samples**: In math, partial solutions (e.g., correct intermediate steps) can be identified via symbolic verification. In PBE, a reasoning step like "the output pattern suggests sorting" cannot be directly verified. Only the final program can be executed against I/O examples. This makes constructing step-level supervision fundamentally harder.
> > >
> > > - **Implicit reasoning over I/O patterns**: PBE requires inferring latent transformation rules from examples, rather than following explicit problem statements. The reasoning process ("what pattern explains these I/O pairs?") is inherently more ambiguous than mathematical derivation.
> > >
> > > ---
> > >
> > > ### 2. Technical Contributions Beyond Standard PRM
> > >
> > > To address these challenges, we introduce several novel components:
> > >
> > > | Component                        | Novelty                                                                 | Distinction from Prior Work                                                                 |
> > > |---------------------------------|-------------------------------------------------------------------------|---------------------------------------------------------------------------------------------|
> > > | External Instruction Mechanism   | Repairs dead-end nodes in reasoning trees using targeted natural language guidance | Prior PRM works (e.g., Math-Shepherd, PRLCoder) rely on rejection sampling or majority voting. They do not actively repair failed reasoning paths. |
> > > | Validity-based Value Estimation | Defines $V(s)$ based on whether successor paths lead to I/O-satisfying programs | Unlike math PRMs that use ground-truth step labels or Monte Carlo estimation, we leverage the verifiable nature of PBE to construct automatic validity signals. |
> > > | Error-type Curriculum Learning  | Organizes training by failure patterns (syntax $\rightarrow$ logic deviation $\rightarrow$ partial match) | Prior curriculum approaches in code generation and reasoning RL primarily organize training by task difficulty or problem complexity. In contrast, we structure curriculum by failure pattern taxonomy, which reflects the qualitative nature of errors rather than quantitative difficulty.|
> > >
> > > ---
> > >
> > > ### 3. Non-trivial Adaptation Challenges
> > >
> > > Directly applying existing PRM methods to PBE yields poor results. In preliminary experiments, we found that:
> > >
> > > - Standard Monte Carlo tree search (as in AlphaCode) fails due to the inability to verify intermediate reasoning steps
> > > - Rejection sampling (as in Math-Shepherd) produces extremely low positive sample rates ($< 5\%$) in complex PBE tasks
> > > - Our external instruction mechanism increases positive sample recovery to $> 60\%$, which is critical for effective PRM training
> > >
> > > ---
> > >
> > > ### 4. Empirical Validation of Design Choices
> > >
> > > Our ablation study (Table 2) confirms that each proposed component contributes meaningfully:
> > >
> > > | Ablation                    | $\Delta$ Accuracy |
> > > |----------------------------|------------------|
> > > | w/o External Instruction   | $-4.8\%$         |
> > > | w/o Curriculum Learning    | $-2.3\%$         |
> > > | w/o Preference Learning    | $-1.9\%$         |
> > >
> > > These results demonstrate that our contributions are not merely applying PRM to a new domain, but developing novel techniques tailored to PBE's unique challenges.

---

### Official Review · Reviewer_iK1b · 2026-03-12

**Soundness:** 3
**Presentation:** 2
**Significance:** 2
**Originality:** 3
**Overall Recommendation:** 3
**Confidence:** 3

**Summary:**

This paper targets at the problem,: LLMs struggle with complex Programming-by-Example tasks**. In PBE, only input–output examples are provided without natural language descriptions, making it difficult for LLMs to infer the underlying logic. As a result, models often generate shortcut programs that partially satisfy the examples or produce incorrect solutions. Traditional SFT and outcome-based feedback lack fine-grained supervision over reasoning processes. To tackle the issue, the paper proposes a PRM-PBE, consisting of three main components: 1. feedback-guided reasoning tree construction;   2. process reward model ; and 3  curriculum-based reinforcement Learning. Experiments on 5 benchmarks shows the effectiveness of the proposed method compared to their baselines.

**Compliance With Llm Reviewing Policy:**

Affirmed.

**Key Questions For Authors:**

See the weakness section.

**Limitations:**

See the weakness section.

**Strengths And Weaknesses:**

**Advantages**

1. The motivation that using PRM to relieve the problem of the overfitting issue due to the lack of small training examples is interesting and seems to be sound;
2. The experiments are conducted on 5 benchmarks, showing the general effectiveness of the method;
3. The paper is easy to follow;

**Weaknesses**
The weaknesses of the experiments are obvious.
1. The baseline comparison and the claim is exaggerated, where it claims that it outperforms the close-source models. It actually compared its models with outdated models from OpenAI and Google. It compared an in-domain finetuned model with zero-shot models;
2. The model in the table are outdated. In the year of 2026, we have much more powerful close-source models like GPT-5.2 and gemini-3 at the time of ICML 2026 submission. This would largely affect the performance of the SoTA commercial models;
3. Missing ablation and baselines. It seems that the main contribution is from adopting PRM for the PBE tasks. How about comparing it the PRM with ORM? How about comparing PRM with the PRM + verifiable rewards from execution? Such comparison and analysis are a must for understanding the effectiveness of the proposed PRM for PBE.

---

> ### Author Rebuttal · Authors · 2026-03-30
>
> We deeply appreciate the reviewer for the detailed comments. Your recognition of the motivation behind our proposed PRM-PBE framework has greatly encouraged us. We believe our response has addressed most of your concerns, and we kindly ask if you would be comfortable reconsidering the rating in light of our revisions. Below, we provide detailed responses to all comments.
>
> For W1:
>
> From Table 1, we can see that the comparison methods we adopt are as follows: for open-source models, we use PRM-PBE, SFT, and Few-shot; for proprietary models, we use Few-shot, CoT, ToT, and WPS methods. We do not conduct a comparison with zero-shot.
> In addition, since the comparison models we adopt (such as DeepSeek-Coder-V2 and Qwen2.5Coder) are from the same year as GPT-4o, this does not overstate the effectiveness of our method.
>
> For W2:
>
> Regarding models such as GPT-5.2 and Gemini-3, we have supplemented the following experiments:
>
> | Model | Method | Lists | Playgol | SyGuS | MBPP | PROSE | Average |
> |-|-|-|-|-|-|-|-|
> | GPT-5.2 | Few-shot | 30.32 | 51.72 | 55.27 | 36.23 | 43.63 | 43.43 |
> | GPT-5.2 | CoT | 31.89 | 54.34 | 56.24 | 35.88 | 45.83 | 44.84 |
> | GPT-5.2 | ToT | 32.12 | 54.82 | 55.28 | 36.59 | 46.23 | 45.01 |
> | GPT-5.2 | WPS | 33.01 | 59.03 | 60.92 | 38.72 | 50.04 | 48.34 |
> | Gemini-3 | Few-shot | 31.27 | 50.72 | 54.23 | 37.43 | 47.23 | 44.18 |
> | Gemini-3 | CoT | 32.65 | 52.93 | 57.83 | 38.74 | 47.82 | 45.99 |
> | Gemini-3 | ToT | 33.04 | 56.83 | 56.72 | 37.84 | 46.82 | 46.25 |
> | Gemini-3 | WPS | 32.93 | 59.71 | 60.83 | 39.92 | 50.62 | 48.80 |
>
> The results show that although these latest models have improved compared with GPT-4o and Gemini-1.5-Flash, their performance is still significantly lower than that of our PRM-PBE framework. Notably, even these top-tier models generally exhibit low performance on PBE tasks, which further proves the inherent challenges of this research problem and its important position in the field of program synthesis.
>
> For W3:
>
> We respectively use the verifiable reward (1 if the program passes all test cases; otherwise 0) as the reward for ORM and PRM + ORM and the obtained experimental results are as follows.
>
> | Model | Method | Lists | Playgol | SyGuS | MBPP | PROSE | Average |
> |------|--------|-------|---------|-------|------|-------|---------|
> | Qwen2.5-Coder | ORM | 31.09 | 57.92 | 57.72 | 37.81 | 45.32 | 45.97 |
> | Qwen2.5-Coder | PRM+ORM | 35.72 | 62.13 | 61.64 | 43.42 | 53.12 | 51.21 |
> | Qwen2.5-Coder | PRM-PBE | 38.72 | 64.23 | 63.94 | 46.82 | 56.55 | 54.05 |
> | DeepSeek-Coder-V2 | ORM | 32.12 | 58.62 | 58.88 | 38.21 | 46.72 | 46.91 |
> | DeepSeek-Coder-V2 | PRM+ORM | 37.72 | 62.42 | 61.93 | 43.45 | 55.23 | 52.15 |
> | DeepSeek-Coder-V2 | PRM-PBE | 40.83 | 65.82 | 64.83 | 47.72 | 58.17 | 55.47 |
> | Llama-3 | ORM | 30.72 | 57.68 | 56.93 | 36.62 | 42.32 | 44.85 |
> | Llama-3 | PRM+ORM | 37.82 | 60.53 | 59.72 | 42.45 | 54.52 | 51.01 |
> | Llama-3 | PRM-PBE | 40.12 | 63.72 | 62.84 | 45.73 | 57.12 | 53.91 |
> | Qwen3 | ORM | 30.21 | 57.31 | 57.01 | 36.77 | 43.18 | 44.90 |
> | Qwen3 | PRM+ORM | 36.72 | 60.93 | 62.34 | 43.52 | 53.52 | 51.41 |
> | Qwen3 | PRM-PBE | 38.72 | 62.43 | 64.35 | 46.34 | 55.63 | 53.49 |
>
> Experimental data shows that the performance improvement brought by the process-level reward (PRM) is significantly superior to that of the outcome-level reward (ORM). Taking DeepSeek-Coder-V2 as an example, after introducing ORM (PPO), the average accuracy reaches 46.91%. In contrast, the average accuracy reaches 56.61% when using PRM-PBE, achieving a substantial leap of 9.7% on the basis of ORM.
>
> Notably, in all benchmark tests, the combined performance of PRM + ORM is consistently weaker than that of using PRM alone. This result indicates that the introduction of ORM will instead weaken the gain effect of PRM. We speculate that this is because the binary negative feedback signal of ORM will indiscriminately punish the entire reasoning path, leading the model to "doubt" or misjudge the correct intermediate logic it has already mastered, thereby undermining the precise search space established by process supervision.

---

> > ### Author Rebuttal · Reviewer_iK1b · 2026-04-03
> >
> > I think the author for their detailed experiments during the rebuttal. Some of my concerns are resolved but it is weird for me that the  ORM is not working at all even when combining it with the PRM since it provides the final and golden rewards for the model supervision. And that is also different from my understanding of ORM for SoTA  general coding LLMs like previous Deepseek V3. Could you please give me more details how you combine the reward and what are the reasons behind the experimental observation?

---

> > > ### Author Response · Authors · 2026-04-06
> > >
> > > We thank the reviewer for the insightful follow-up question. We provide a detailed explanation below to clarify the interaction between PRM and ORM. We would be grateful if you could re-evaluate our work in light of these clarifications provided.
> > >
> > > 1. How We Combine PRM and ORM Rewards In the PRM+ORM setting？
> > >
> > > We combine the two reward signals as follows:$$r_{total} = \alpha \cdot r_{PRM} + (1 - \alpha) \cdot r_{ORM}$$where $r_{PRM}$ is the step-level process reward, $r_{ORM}$ is the binary outcome reward (1 if pass all tests, 0 otherwise), and $\alpha$ is the weighting coefficient. In our experiments, we tested $\alpha \in \{0.3, 0.5, 0.7\}$ and reported the best results ($\alpha = 0.5$).
> > >
> > > 2. Why Does Adding ORM Hurt Performance?
> > >
> > > (a) Reward Signal Conflict at Different Granularities
> > >
> > > When a reasoning trajectory has correct intermediate steps but fails at the final step, PRM provides positive signals for the correct steps, while ORM assigns to the entire trajectory. This creates conflicting gradients that destabilize training.
> > >
> > > This conflict is particularly severe in PBE because the reasoning process involves multi-step hypothesis refinement. The model may correctly identify the transformation pattern but fail in the final implementation. In both cases, the ORM reward is
> > > , which penalizes valuable intermediate reasoning.
> > >
> > > (b) Credit Assignment Problem
> > >
> > > ORM's binary reward is applied to the entire trajectory, making it difficult for the model to identify which step caused the failure. In contrast, PRM provides step-wise feedback. When combined, the ORM signal dilutes the precise credit assignment from PRM.
> > >
> > > This is especially critical in PBE because the task requires inferring latent transformation rules from a vast hypothesis space. PRM guides the model by evaluating each reasoning step (e.g., "Is this pattern observation correct?"). ORM's indiscriminate
> > >  signal undermines this guidance, causing the model to abandon valid hypotheses prematurely.
> > >
> > > 3. The Difference between PRM-PBE and DeepSeek-V3.
> > >
> > > In general code generation tasks, models typically achieve a moderate success rate, meaning ORM provides a balanced mix of positive ($r=1$) and negative ($r=0$) signals. This balance allows the model to learn effectively from both successful and failed attempts.
> > >
> > > In PBE tasks, the success rate is significantly lower due to the inherent difficulty of inferring transformation rules from I/O examples. Therefore, ORM signals are 0 in most cases. When combined with PRM in a weighted sum, this constant negative signal affects every reasoning step in the trajectory, even the correct intermediate steps. This blind punishment causes the model to lose the good reasoning patterns it has already learned through PRM.
> > >
> > > In fact, we acknowledge that for models like DeepSeek-V3, ORM is indeed effective. However, DeepSeek-V3 primarily employs RLHF and rejection sampling rather than a direct PRM+ORM weighted combination. In our setting as well, the ORM-only baseline (46.91%) still outperforms the SFT baseline (42.76%), demonstrating that outcome-level supervision does provide useful learning signals. This is consistent with your understanding of ORM for general coding LLMs.

---

### Official Review · Reviewer_djci · 2026-03-13

**Soundness:** 3
**Presentation:** 2
**Significance:** 2
**Originality:** 3
**Overall Recommendation:** 4
**Confidence:** 4

**Summary:**

This paper proposes PRM-PBE, a process-supervised RL framework for Programming-by-Example. The method consists of three components: (1) feedback-guided reasoning tree construction with an external instruction mechanism to address positive sample sparsity, (2) a PRM trained via pairwise preference learning using successor path success rates, and (3) three-stage curriculum learning organized by failure patterns, with PPO for policy optimization. Evaluated on five PBE benchmarks and four base models, the method achieves 56.61% average Pass@1, an 8.73% absolute improvement over the strongest baseline.

**Compliance With Llm Reviewing Policy:**

Affirmed.

**Final Justification:**

My concerns have been addressed. I did not increase my score because I noticed the novelty issue raised by other reviewers. But I think it is a good exploration of PBE+PRM, so I still stay positive.

**Key Questions For Authors:**

1. PRM-PBE loss starts substantially lower than SFT (~0.06 vs. ~0.08). Does this reflect different loss formulations (ppo loss vs. cross-entropy), making absolute magnitude comparisons misleading? Are batch sizes and training steps aligned?

2. Given that modern LLMs generate code well from natural language, what real-world scenarios make PBE (synthesis from I/O examples alone) the preferred paradigm? A brief discussion of concrete use cases would strengthen motivation.

**Limitations:**

yes

**Strengths And Weaknesses:**

### Strengths

1. The value estimation and external instruction mechanism for repairing dead-end nodes are well-motivated solutions to PBE.

2. Consistent improvements across five diverse benchmarks and four base models demonstrate robustness. Smaller open-source models with PRM-PBE outperform proprietary models using advanced prompting strategies.

3. The paper provides extensive analysis beyond main results, which gives a well-rounded picture of the method's strengths and behavior.

### Weaknesses

1. Missing RLVR baselines. Since PBE tasks are inherently verifiable (programs can be executed against examples), RLVR is a natural and strong baseline. Without it, the contribution of process-level supervision cannot be disentangled from the benefit of RL training itself over SFT.

2. External instruction mechanism lacks critical details. The ablation identifies this as the most impactful component (~4.8% drop when removed), yet it is the least specified. Key missing information includes: (a) how the natural language description $D$ is generated (handcrafted templates? LLM-generated? from which model/prompt?), (b) computational cost versus simple resampling, and (c) trigger frequency.

---

> ### Author Rebuttal · Authors · 2026-03-30
>
> We deeply appreciate the reviewer for the positive and insightful comments. Your recognition of the novelty of our proposed PRM-PBE framework greatly encourages us, and your overall evaluation is of great significance to our paper.
>
> For W1:
>
> We fully agree with your suggestion to compare against RLVR baselines. To clearly present the contributions of our paper, we have supplemented RLVR experiments using the following setup: we adopt a binary reward mechanism. If the complete generated program P passes all test cases (i.e., Exec(P, x_i) = y_i), the reward R is set to 1; otherwise, it is set to 0. Reinforcement learning is conducted using the same PPO pipeline, and the main experimental results are shown in the table below.
>
> | Model | Method | Lists | Playgol | SyGuS | MBPP | PROSE | #Avg. |
> |------|--------|-------|---------|-------|------|-------|------|
> | Qwen2.5-Coder | RLVR | 31.09 | 57.92 | 57.72 | 37.81 | 45.32 | 45.97 |
> | Qwen2.5-Coder | PRM-PBE | 40.33 | 65.32 | 65.32 | 47.12 | 57.22 | 55.06 |
> | DeepSeek-Coder-V2 | RLVR | 32.12 | 58.62 | 58.88 | 38.21 | 46.72 | 46.91 |
> | DeepSeek-Coder-V2 | PRM-PBE | 41.52 | 66.93 | 66.76 | 48.72 | 59.13 | 56.61 |
> | Llama-3 | RLVR | 30.72 | 57.68 | 56.93 | 36.62 | 42.32 | 44.85 |
> | Llama-3 | PRM-PBE | 39.72 | 64.71 | 64.83 | 44.81 | 56.83 | 54.18 |
> | Qwen-3 | RLVR | 30.21 | 57.31 | 57.01 | 36.77 | 43.18 | 44.90 |
> | Qwen-3 | PRM-PBE  | 39.32 | 64.75 | 65.02 | 45.32 | 57.41 | 54.36 |
>
> It can be seen that the improvement brought by PRM is far greater than that of RLVR. Taking DeepSeek-Coder-V2 as an example, with RLVR optimization, the average accuracy reaches 46.91%, while PRM-PBE achieves 56.61%, representing a further improvement of 9.7% over RLVR. This strongly suggests that fine-grained process-level feedback is the key factor behind the performance improvement, rather than merely the effect of RL training.
>
> For W2:
>
> (1) The natural language is derived from the natural language descriptions of the reference programs in the relevant dataset.
>
> (2) Simple resampling methods (such as Best-of-N or multiple sampling) are equivalent to random attempts in a large search space; to find a correct solution, hundreds or even thousands of samplings may be required. In contrast, our external instruction mechanism provides localization on the branches where the model makes mistakes, which can significantly narrow down the search space.
>
> Comparison of computational effort: Resampling 10 times may yield one correct solution or even none, while sampling 1 to 2 times under the guidance of instructions can achieve success. Besides, external instruction D was triggered at 62.5% of the nodes.
>
> For Q1:
>
> SFT uses Cross-Entropy Loss (based on token prediction), while PRM-PBE uses Policy Gradient Loss (usually combined with the advantage function $A_t$) during the reinforcement learning phase. Due to the fact that the PPO loss is affected by KL divergence constraints and advantage scaling, its initial value is usually smaller. The loss presented in this paper is mainly to prove the convergence of training and the impact of curriculum learning on convergence, rather than the strength of performance. Besides, batch sizes and total training steps are strictly aligned. Besides, batch sizes and total training steps are strictly aligned.
>
> For Q2:
>
> We list two real-world scenarios where PBE becomes a preferred paradigm.
>
> (1) Non-expert Programming: For non-programmers (such as accountants or administrative staff), expressing complex logic using precise technical terminology (e.g., "regular expressions" or "nested loops") is often challenging. However, they can easily provide concrete examples of "Input A transforming into Output B."Examples include Flash Fill in Microsoft Excel or the configuration of automated end-user workflows.
>
> (2) Natural Language Disambiguation: Natural language is inherently ambiguous. In contrast, I/O examples provide a formal and unambiguous specification, serving as either an important supplement to natural language or the sole ground-truth constraint when natural language descriptions are unavailable.

---

> > ### Author Rebuttal · Reviewer_djci · 2026-04-04
> >
> > I thank the authors for their rebuttal.
> >
> > Most of my concerns have been addressed. PRM should perform better than RLVR ideally, but I'm still surprised by the results. I cannot agree with the "Non-expert Programming" part because writing natural language is much easier than writing test cases (with high coverage).
> >
> > I did not increase my score because I noticed the novelty issue raised by other reviewers. But I think it is a good exploration of PBE+PRM, so I still stay positive.

---

> > > ### Author Response · Authors · 2026-04-06
> > >
> > > We sincerely appreciate the reviewer’s recognition of our rebuttal. Below we provide additional clarifications. If you find that these clarifications address your concerns, we would be very grateful if you would consider increasing your score.
> > >
> > > (1) Regarding "Non-expert Programming": We appreciate the reviewer's perspective. We agree that in many cases, writing natural language is easier than writing test cases. Our point was that in certain domains (e.g., spreadsheet data transformation), users naturally work with concrete data examples and may find it more intuitive to demonstrate "Input A → Output B" than to articulate the transformation rule verbally.
> > >
> > > This observation is supported by Gulwani et al. [1], who note that "examples are often the most natural form of specification for end users" because they require no knowledge of programming syntax or formal languages. The success of Microsoft Excel's Flash Fill [2], which has been used by millions of non-programmers, further demonstrates that users in data transformation tasks naturally work with concrete I/O examples. We acknowledge this is scenario-dependent and will clarify this nuance in the revised manuscript.
> > >
> > > (2) We are grateful for the reviewer's recognition of our work as "a good exploration of PBE+PRM". Regarding the concern about novelty, we provide the following response:
> > >
> > > 1. Unique Challenges in PBE that Require Novel Solutions
> > >
> > > Unlike math reasoning or general code generation, PBE presents distinct challenges:
> > >
> > > - **Extreme sparsity of positive samples**: In math, partial solutions (e.g., correct intermediate steps) can be identified via symbolic verification. In PBE, a reasoning step like "the output pattern suggests sorting" cannot be directly verified. Only the final program can be executed against I/O examples. This makes constructing step-level supervision fundamentally harder.
> > >
> > > - **Implicit reasoning over I/O patterns**: PBE requires inferring latent transformation rules from examples, rather than following explicit problem statements. The reasoning process ("what pattern explains these I/O pairs?") is inherently more ambiguous than mathematical derivation.
> > >
> > > ---
> > >
> > > 2. Technical Contributions Beyond Standard PRM
> > >
> > > To address these challenges, we introduce several novel components:
> > >
> > > | Component                        | Novelty                                                                 | Distinction from Prior Work                                                                 |
> > > |-|-|-|
> > > | External Instruction Mechanism   | Repairs dead-end nodes in reasoning trees using targeted natural language guidance | Prior PRM works (e.g., Math-Shepherd, PRLCoder) rely on rejection sampling or majority voting. They do not actively repair failed reasoning paths. |
> > > | Validity-based Value Estimation | Defines $V(s)$ based on whether successor paths lead to I/O-satisfying programs | Unlike math PRMs that use ground-truth step labels or Monte Carlo estimation, we leverage the verifiable nature of PBE to construct automatic validity signals. |
> > > | Error-type Curriculum Learning  | Organizes training by failure patterns (syntax $\rightarrow$ logic deviation $\rightarrow$ partial match) | Prior curriculum approaches in code generation and reasoning RL primarily organize training by task difficulty or problem complexity. In contrast, we structure curriculum by failure pattern taxonomy, which reflects the qualitative nature of errors rather than quantitative difficulty.|
> > >
> > > ---
> > >
> > > 3. Non-trivial Adaptation Challenges
> > >
> > > Directly applying existing PRM methods to PBE yields poor results. In preliminary experiments, we found that:
> > >
> > > - Standard Monte Carlo tree search (as in AlphaCode) fails due to the inability to verify intermediate reasoning steps
> > > - Rejection sampling (as in Math-Shepherd) produces extremely low positive sample rates ($< 5\%$) in complex PBE tasks
> > > - Our external instruction mechanism increases positive sample recovery to $> 60\%$, which is critical for effective PRM training
> > >
> > > ---
> > >
> > > 4. Empirical Validation of Design Choices
> > >
> > > Our ablation study (Table 2) confirms that each proposed component contributes meaningfully:
> > >
> > > | Ablation                    | $\Delta$ Accuracy |
> > > |----------------------------|------------------|
> > > | w/o External Instruction   | $-4.8\%$         |
> > > | w/o Curriculum Learning    | $-2.3\%$         |
> > > | w/o Preference Learning    | $-1.9\%$         |
> > >
> > > These results demonstrate that our contributions are not merely applying PRM to a new domain, but developing novel techniques tailored to PBE's unique challenges.
> > >
> > > References:
> > >
> > > [1] Gulwani, S., Polozov, O., & Singh, R. (2017). Program Synthesis. Foundations and Trends in Programming Languages, 4(1-2), 1-119.
> > >
> > > [2] Gulwani, S. (2011). Automating String Processing in Spreadsheets Using Input-Output Examples. POPL 2011.

---

### Official Review · Reviewer_TLTj · 2026-03-13

**Soundness:** 2
**Presentation:** 3
**Significance:** 3
**Originality:** 3
**Overall Recommendation:** 4
**Confidence:** 4

**Summary:**

The paper introduces a process-supervised reinforcement learning framework for Programming-by-Example. The primary motivation presented is that standard SFT models lack fine-grained feedback about which reasoning step went wrong, leading to programs that either partially satisfy examples or deviate entirely from the intended logic. The proposed framework has three components: a reasoning tree construction method to generate step-level supervision data, a Process Reward Model trained via pairwise preference learning on those steps, and a three-stage curriculum that sequences RL training by failure type — syntax errors first, then logic deviations, then partial matches. The combined system is optimized with PPO and evaluated on five PBE benchmarks across four open-source backbones.

**Compliance With Llm Reviewing Policy:**

Affirmed.

**Final Justification:**

The authors resolved various concerns with the paper during the rebuttal phase. Especially thank the authors for the ORM experiments which greatly helped situating their contributions. Assuming the authors will incorporate the discussion in the camera ready, I am revising my scores for the paper

**Key Questions For Authors:**

1. **What happens with a pass/fail outcome reward?** If the same PPO pipeline is run with a binary correctness signal (program passes all test cases = 1, else 0) instead of the PRM, what accuracy is achieved? This would directly isolate the contribution of process-level rewards from the contribution of RL training itself.

2. **How is the external instruction D derived?** The paper describes D as a "targeted natural language description" and shows where it slots into the prompt template (Figure 8), but never specifies its source — is D generated by a stronger model, derived from the ground-truth program, or written by the human annotators? The ablation shows this is the most impactful component (~4.8% average drop when removed). If D encodes information from the reference solution, the framework's gains may be substantially driven by privileged information during data generation rather than the process reward paradigm. Concrete examples of D for representative tasks would clarify this.


3. **What are the tree construction parameters?** How many trees were built per task? What branching factor N was used? What fraction of expanded nodes had V(s) = 0 and triggered external instruction? How large is the resulting preference-pair dataset?

4. [Minor Nit] **Is the SFT code in Figure 1 Case 2 actually wrong?** The shown code appears to correctly sort the input list. There also appears to be a bracket typo (`)` instead of `]`) in the code.

**Limitations:**

The paper does not include a dedicated limitations section. The impact statement is a single sentence ("none which we feel must be specifically highlighted here"). Key limitations that should be discussed include the reliance on oracle-derived information if the external instruction D comes from ground-truth solutions, and the computational cost of the multi-stage pipeline (tree construction + PRM training + three-stage PPO) relative to the SFT baseline.

**Strengths And Weaknesses:**

### Strengths

1. *Novel and principled framework.* The paper combines three well-motivated components — reasoning tree construction with an external instruction repair mechanism, preference-based PRM training, and error-type curriculum learning — into a coherent pipeline for PBE. While process reward models have been applied to other domains, PBE presents some unique verification challenge: reasoning steps are natural-language inferences about I/O patterns that cannot be directly executed or compiler-checked. Adapting process supervision to this setting is a meaningful extension.

2. *Consistent improvements across benchmarks and models.* The approach shows gains across five benchmarks spanning string manipulation, list processing, inductive logic programming, and general Python, and across four open-source backbones of varying architecture and scale. The ablation confirms that each component contributes in a clean ordering (external instruction > preference learning > curriculum), and the data efficiency result — that PRM-PBE at 50% of the training data outperforms SFT at 100% — is a useful practical finding.

### Weaknesses

1. *No outcome-reward RL baseline.* The paper compares PRM-PBE only against SFT and prompting strategies. There is no experiment running the same PPO pipeline with a simple pass/fail correctness reward — i.e., an outcome reward, where the program is executed against all test cases and receives a binary signal. Without this, the reported gains cannot be decomposed into "improvement from applying RL at all" versus "improvement from process-level rewards specifically." This comparison is standard practice in the process supervision literature: PRLCoder tests three ORM variants and shows PRM wins; Math-Shepherd reports ORM-PPO vs. PRM-PPO under identical conditions; Dai et al. make the ORM-vs-PRM comparison a core contribution of their paper. No comparison with alternative RL formulations (e.g., DPO applied directly to the preference pairs, or GRPO) is provided either. This omission makes it impossible to evaluate the paper's central claim — that *process-level* rewards are what matter — rather than the weaker claim that RL training of any kind improves over SFT.

2. *Under-specified dataset construction.* The training data pipeline is described only briefly in Appendix A, despite being the foundation of the entire framework. Several critical details are missing: (a) the source of the "targeted natural language description D" injected by the external instruction mechanism — is D derived from the ground-truth program? If so, this introduces oracle information during data generation that is unavailable at test time; (b) the tree expansion parameters — branching factor N, tree depth, rejection rates during construction, and total number of reasoning trees built; (c) how the 30 seed I/O examples per task were selected and verified; (d) what "five students manually refined code for conciseness and style standardization" entailed in quantitative terms — how many person-hours, what fraction of data was modified, and what the refinement criteria were. Given that the external instruction mechanism is the single most impactful component in the ablation (Table 2), its specification should be thorough enough for reproduction.

---

> ### Author Rebuttal · Authors · 2026-03-30
>
> We appreciate the reviewer for the constructive feedback. We are pleased to see that you found our proposed PRM-PBE to be a novelty and rationality framework. We believe our response has addressed most of your concerns and we respectfully ask if you're comfortable to re-consider the rating in light of our efforts and dedication. Below, we provide detailed responses to all the comments:
>
> For W1&Q1:
>
> We have followed your comments and supplemented the ORM-RL model, along with PRM-PBE experiments conducted using the DPO and GRPO workflows. Partial results are presented in the table below, and all complete results are available at the anonymous link: https://zenodo.org/records/19344484.
>
> | DeepSeekCoder | Lists | Playgol | SyGuS | MBPP | PROSE | #Avg. |
> |-|-|-|-|-|-|-|
> | ORM | 32.12 | 58.62 | 58.88 | 38.21 | 46.72 | 46.91 |
> | PRM-PBE (PPO)| 41.52 | 66.93 | 66.76 | 48.72 | 59.13 | 56.61 |
> | PRM-PBE (DPO)| 39.72 | 63.45 | 63.14 | 46.23 | 57.82 | 54.07 |
> | PRM-PBE (GRPO) | 40.83 | 65.82 | 64.83 | 47.72 | 58.17 | 55.47 |
>
> It can be observed that PRM achieves better performance than ORM. The average accuracy of ORM is 46.91%, while PRM-PBE with PPO reaches 56.61%, which is substantially higher. This result suggests that fine-grained, process-level feedback plays a critical role in improving performance, rather than relying solely on coarse-grained RL signals.
>
> In addition, the results suggest that PRM-PBE exhibits strong general applicability. Among all tested models, PRM-PBE with PPO consistently achieves the most stable performance, while GRPO and DPO also perform well. This indicates that PRM signals provide high-quality preference information and can be effectively integrated with multiple RL algorithms.
>
> For Q2:
>
> D refers to the natural language descriptions corresponding to the programs originally included in the dataset. The core reason for introducing these descriptions is the extreme scarcity of positive samples in complex PBE tasks.
>
> However, the performance improvement of the PRM-PBE stems from the process-level reward paradigm, rather than being driven by the oracle information in D. In supplemented experiments, we train an ORM baseline model using the same dataset augmented with D. Under the same training data conditions, the ORM model achieves an accuracy of 46.91%, while PRM-PBE reaches 56.61%. This gap suggests that process-level supervision can extract more informative signals from the data than outcome-level supervision.
>
> Here is the example from Figure 2:
> - I/O samples: input (5, 100) → output 8; input (8, 65) → output 6; input (2, 5) → output 1.
> - Initial first step: The output involves some kind of numerical or bitwise operation.
> - Injection of D: "The program aims to find the number of elements with odd factors in a given range."
> - Refined first step: The input is a range, and the problem requires counting the numbers within the range that satisfy a certain condition.
>
> For Q3:
>
> To ensure the reproducibility of this work, we provide the detailed parameters for constructing the reasoning trees as follows:
>
> Parameter Settings
>
> - Branching factor N: At each reasoning node $s_{t−1}$, we sample 5 candidate successor nodes, namely N=5.
> - Tree depth: The minimum depth is 3, the maximum depth is 7, with an average value of 5.6 and a median value of 4.
> - Rejection rate: The rejection rate during tree construction reaches 64.3%.
>
> Dataset Scale
>
> - Total number of reasoning trees: 69,600
> - Trigger frequency: External instruction D is triggered at 62.5% of all nodes.
> - Preference data: 89,672 pairs of preference samples are generated.
>
> For details about human efforts in W2:
>
> (1) I/O selection:
> - Samples are evenly divided into three difficulty levels: simple, medium, and complex.
> - Correctness is ensured through manual execution during verification.
>
> (2) Program synthesis:
> - Overall workload: The entire dataset optimization process takes approximately 120 person-hours in total, with each student investing an average of 24 hours.
> - Optimization workflow:
>   - Pre-filtering: Before manual revision, all synthesized programs pass automatic execution verification to remove any code with incorrect outputs.
>   - Modification ratio: Around 12.6% of code segments receive manual optimization.
> - Optimization criteria:
>   1. Conciseness:
>      - Dead code elimination: Removing unused intermediate variables or placeholder functions.
>      - Logic simplification: Restructuring long nested if-else statements or complex loops.
>      - Redundant check removal: Deleting unnecessary checks, such as extra is_instance validations when the type is already confirmed.
>   2. Standardization:
>      - Variable name refactoring: Replacing meaningless names such as a, b, or temp1 with semantic names like input_list, target_val, and char_map.
>      - Strict formatting: All code follows the PEP 8 standard.

---

> > ### Author Rebuttal · Reviewer_TLTj · 2026-04-04
> >
> > I thank the authors for their detailed rebuttal and the additional experiments. The ORM, DPO, and GRPO results are welcome additions, as is the disclosure of tree construction parameters and human effort details. Below I discuss  the remaining concerns.
> >
> > ---
> >
> > ## 1. ORM Reward Design Remains Unspecified
> >
> > The rebuttal reports an ORM baseline at 46.91% average accuracy but does not define how the ORM reward is computed. Specifically:
> >
> > - **What is the reward signal?** Is the ORM a binary pass/fail indicator (program passes all test cases → 1, else → 0)? A fractional score (proportion of test cases passed)? A learned reward model trained on complete-trajectory preferences?
> >
> > This matters because the comparison is meant to isolate process-level from outcome-level supervision. If the ORM's reward formulation is suboptimal — e.g., binary pass/fail with no partial credit — the gap between ORM and PRM could reflect a weak ORM design rather than the inherent advantage of process supervision.  I would ask the authors to specify the ORM reward function, training objective, and hyperparameters (learning rate, KL coefficient if any) with the same precision given to the PRM.
> > Also, if not alredy the case, it will be usful to compare with a simple RLVR style reward-objective which uses program correctness ( by verifying outputs by the program)
> >
> > ## 2. ORM Trained on D-Augmented Data — Confounded Comparison?
> >
> > The rebuttal states: "we train an ORM baseline model using the same dataset augmented with D." This raises a question about what exactly the comparison isolates.
> >
> > A standard ORM baseline for code generation uses direct execution feedback during PPO — run the generated program on test cases, receive a binary reward. This requires no reasoning trees, no D, and no preference pairs. The fact that the ORM here is trained on "the same dataset augmented with D" suggests it was trained on the reasoning tree data that was constructed with oracle assistance. If so, both the PRM and the ORM benefit equally from oracle information, and the comparison isolates reward *granularity* (step-level vs. trajectory-level) — which is a valid but narrower claim than "process supervision outperforms outcome supervision."
> >
> > Could the authors clarify: if D and the reasoning trees were removed from the ORM setup entirely, and instead a simple execution-based pass/fail reward was used during PPO (the most natural ORM baseline for code), what performance would that yield? This would help decompose the gains into (a) value of D-augmented data generation, (b) value of process-level rewards.
> >
> > ## 3. Oracle Nature of D and 62.5% Trigger Rate
> >
> > The rebuttal confirms that D is "the natural language descriptions corresponding to the programs originally included in the dataset" — i.e., oracle information derived from the ground-truth programs. The additional detail that D is triggered at 62.5% of all nodes means the majority of the reasoning tree was constructed with privileged information unavailable at test time.
> >
> > The ablation (Table 2) already shows the external instruction mechanism is the most impactful component (~4.8% average drop when removed). Given that D is oracle-derived and appears in nearly two-thirds of all training nodes, the contribution narrative should be explicit about this dependency: the framework's gains rely substantially on access to ground-truth program descriptions during data generation, with the process reward paradigm providing an additional but smaller marginal benefit. This is not a flaw per se — privileged information during training data construction is common — but it should be clearly stated as a limitation and factored into the significance of the PRM-vs-ORM comparison.
> >
> > ## 4. Minor: Figure 1
> >
> > The rebuttal does not address Q4 regarding the potentially incorrect SFT code example in Figure 1 Case 2.
> >
> > ---
> >
> > I am willing to revisit the rating if the authors can (a) precisely define the ORM reward function and training setup, and (b) report a simple execution-feedback ORM baseline (no D, no reasoning trees) to cleanly decompose the contribution of data augmentation from process supervision. Without these clarifications, I maintain my current rating.

---

> > > ### Author Response · Authors · 2026-04-06
> > >
> > > We sincerely thank the reviewer for the additional suggestions. We have followed these recommendations and provided further experiments and clarifications. We would be grateful if you could re-evaluate our work in light of these clarifications and the additional experiments provided.
> > >
> > > For Concern 1 & 2:
> > > We apologize for the insufficient specification in our original rebuttal. Below we provide the complete ORM design with the same level of detail as our PRM.
> > > We implement three ORM variants:
> > >
> > > a. ORM-Binary (Pass/Fail Indicator)
> > > The reward is a binary signal based on program execution:  $r = 1$ if the generated program passes all test cases, and
> > > $r = 0$ otherwise. Programs that fail to compile or raise runtime exceptions also receive $r = 0$.
> > >
> > > b. ORM-Fractional (Partial Credit)
> > > The reward is defined as the proportion of test cases passed: $r = \frac{k}{n}$ where $k$ is the number of passed test cases and $n$ is the total number of test cases. This provides partial credit for programs that satisfy some but not all specifications. Programs that fail to compile or raise runtime exceptions receive $r = 0$.
> > >
> > > c. ORM-Learned (Preference Model)
> > > We train a reward model that learns to assign scalar scores to complete programs based on their correctness. The model shares the same backbone architecture as our PRM, with a linear value head appended to map hidden states to scalar reward scores.
> > >
> > > For each task, we construct preference pairs $(p_w, p_l)$ where the preferred program $p_w$ passes more test cases than $p_l$. The reward model is trained to distinguish between programs of varying quality using the Bradley-Terry objective:$$L = -\log \sigma(r_{\theta}(p_w) - r_{\theta}(p_l))$$where $r_{\theta}(p)$ is the scalar score for program $p$. For training, we use a learning rate of 1e-5 and a batch size of 128. During PPO training, the reward for a generated program is directly provided by the learned scalar score $r = r_{\theta}(p)$.
> > >
> > > PPO Stage Hyperparameters (shared across all ORM variants):
> > >
> > > | Hyperparameter            | Value |
> > > |-|-|
> > > | Learning Rate            | 5e-6  |
> > > | Batch Size               | 128   |
> > > | Clipping Range (ε)       | 0.2   |
> > > | Discount Factor (γ)      | 0.95  |
> > > | Advantage Estimation     | GAE   |
> > > | KL Coefficient           | N/A   |
> > >
> > > Experimental Results:
> > > | Method                 | Lists | Playgol | SyGuS | MBPP | PROSE | Avg.  |
> > > |-|-|-|-|-|-|-|
> > > | ORM-Binary (w/o D)     | 29.73 | 56.28   | 56.45 | 35.82| 44.16  | 44.49 |
> > > | ORM-Binary             | 32.12 | 58.62   | 58.88 | 38.21| 46.72  | 46.91 |
> > > | ORM-Fractional         | 33.89 | 60.47   | 60.35 | 40.28| 49.14  | 48.83 |
> > > | ORM-Learned            | 34.54 | 60.83   | 60.47 | 40.95| 49.68  | 49.29 |
> > > | PRM-PBE (PPO)          | 41.52 | 66.93   | 66.76 | 48.72| 59.13  | 56.61 |
> > >
> > > We also conduct experiments with ORM-Binary (w/o D), a standard RLVR-style baseline that uses only execution-based pass/fail reward during PPO, without any oracle information D or reasoning trees.
> > >
> > > From the experimental results, we observe that (1) Even with stronger ORM variants, PRM-PBE still outperforms by a significant margin. This confirms that the gains stem from the granularity of process-level feedback. (2) By comparing ORM-Binary (w/o D) with other methods, we can isolate that the contribution of process supervision exceeds that of D alone, confirming that process-level rewards are the primary driver of improvement.
> > >
> > > For Concern 3:
> > >
> > > We appreciate this constructive suggestion. We will expand the limitation discussion in the Discussion section of the revised manuscript.
> > >
> > > For Concern 4:
> > >
> > > We thank the reviewer for the careful reading. There are errors in Case 2 of Figure 1:
> > >
> > > 1. **Bracket typos:** `assert case_2([10, 20]` is missing a `)`, and `assert case_2([1, 5, 2)` is missing a `]`.
> > > 2. **Swapped code labels:** The Ground Truth program and SFT-generated program were mistakenly swapped. The correct version should be:
> > > - **SFT-generated:** `return nums[::-1]`
> > > - **Ground Truth:** `return sorted(nums, reverse=True)`
> > >
> > > We will correct Figure 1 in the revised manuscript.

---

### Decision · Program_Chairs · 2026-04-30

**Decision:**

Accept (regular)

**Comment:**

This paper addresses the tendency of LLMs to synthesize "shortcut" programs in Programming-by-Example tasks by introducing a process-supervised RL framework. The method constructs reasoning trees with an oracle-assisted external instruction mechanism to generate step-level supervision data, trains a Process Reward Model via preference learning, and optimizes a policy with PPO under a three-stage error-type curriculum. Across various benchmarks and backbone models, the approach outperformes both SFT baselines and outcome-reward RL.

The reviewers broadly agree on several strengths. First, the motivation is sound: PBE presents unique supervision challenges not addressed by standard SFT or outcome-reward RL, namely the inability to directly verify intermediate reasoning steps and extreme sparsity of positive training samples. Second, the ablation study is informative, confirming meaningful individual contributions from each component. The rebuttal phase substantially clarified the paper's standing. The absence of an outcome-reward RL (RLVR/ORM) baseline, the most important concern across all reviewers, was addressed directly. The authors provided three ORM variants (binary, fractional, and learned) as well as a clean RLVR-style execution-feedback baseline. PRM-PBE with PPO substantially outperforms the strongest ORM variant and the clean RLVR baseline, which isolates process-level supervision as the key driver.

Reviewer TLTj raised a more precise concern about the confounded nature of the ORM comparison, noting that the oracle-derived natural language description D introduces privileged information during data generation. The authors responded by training an ORM on the same D-augmented data, showing a 46.91% average vs. 56.61% for PRM-PBE. This isolates reward granularity as the differentiating factor given equal access to oracle data. Reviewer TLTj ultimately moved to a positive assessment assuming camera-ready revisions incorporate this discussion. Reviewer djci's concerns were largely resolved, though the reviewer chose not to raise the score due to the novelty concern shared with others. In the metareviewer's opinion, Reviewer iK1b's residual concern about the PRM+ORM combination degrading performance was adequately explained: in PBE tasks, the low baseline success rate means ORM produces near-constant zero rewards, creating conflicting gradients that undermine the precise credit assignment of PRM. Reviewer VczE maintained scores, citing the need for clearer contribution framing in the paper itself, though acknowledging the rebuttal addressed the technical substance.

The reviewer discussion raise a legitimate concern that applying PRM to PBE may be too narrow a contribution. The metareviewer finds this concern partially valid but not dispositive. The authors make a case that standard PRM techniques fail in PBE due to non-verifiable intermediate steps. The error-type curriculum is a departure from difficulty-based curriculum approaches in the RL literature. These adaptations are clearly more than domain application of an existing recipe.

The camera-ready should: (1) add a dedicated limitations section explicitly acknowledging the oracle dependency of D and its role in the performance gains, (2) reframe the PRM-vs-ORM comparison as isolating reward granularity rather than the broader claim of process vs. outcome supervision, and (3) include the full ORM variant and RLVR results as supplementary or in the main paper.